# Targeted attenuation of elevated histone marks at *SNCA* alleviates α-synuclein in Parkinson's disease

Subhrangshu Guhathakurta[1], Jinil Kim[1,2], Levi Adams[1,3], Sambuddha Basu[1], Min Kyung Song[1,3], Evan Adler[1], Goun Je[1], Mariana Bernardo Fiadeiro[1] & Yoon-Seong Kim[1,3,*] (iD)

## Abstract

Epigenetic deregulation of α-synuclein plays a key role in Parkinson's disease (PD). Analysis of the *SNCA* promoter using the ENCODE database revealed the presence of important histone post-translational modifications (PTMs) including transcription-promoting marks, H3K4me3 and H3K27ac, and repressive mark, H3K27me3. We investigated these histone marks in post-mortem brains of controls and PD patients and observed that only H3K4me3 was significantly elevated at the *SNCA* promoter of the substantia nigra (SN) of PD patients both in punch biopsy and in NeuN-positive neuronal nuclei samples. To understand the importance of H3K4me3 in regulation of α-synuclein, we developed CRISPR/dCas9-based locus-specific H3K4me3 demethylating system where the catalytic domain of JARID1A was recruited to the *SNCA* promoter. This CRISPR/dCas9 SunTag-JARID1A significantly reduced H3K4me3 at *SNCA* promoter and concomitantly decreased α-synuclein both in the neuronal cell line SH-SY5Y and idiopathic PD-iPSC derived dopaminergic neurons. In sum, this study indicates that α-synuclein expression in PD is controlled by *SNCA*'s histone PTMs and modulation of the histone landscape of *SNCA* can reduce α-synuclein expression.

**Keywords** CRISPR/Cas9; histone post-translational modifications; iPSCs; Parkinson's disease; α-synuclein

**Subject Categories** Chromatin, Transcription & Genomics; Neuroscience

## Introduction

Parkinson's disease (PD) is the second most prevalent neurodegenerative disease affecting nearly one million people worldwide. In the USA alone, 60,000 patients are diagnosed with PD each year (Parkinson's Foundation). PD is a late-onset disease that destroys 70–80% of dopaminergic neurons in the substantia nigra pars compacta region of the midbrain before motor symptoms are typically noticeable (Morrish *et al*, 1998; Postuma *et al*, 2010; Heisters, 2011). The disease is usually idiopathic, but cases with known genetic components account for around 10% of reported cases (Gasser, 2009). Of note, α-synuclein is one of the primary proteins linked with PD, and it has been identified as playing a role in both genetic and non-genetic cases.

The first evidence of α-synuclein involvement in PD came from several familial studies that showed individuals harboring coding region mutations in *SNCA*, and the gene encoding α-synuclein, including A53T, E46K, and A30P, had early-onset disease with an autosomal dominant pattern of inheritance (Polymeropoulos *et al*, 1997; Kruger *et al*, 1999; Zarranz *et al*, 2004). Later, more familial *SNCA* mutations were identified including H50Q, G51D, A18T, and A29S, all of which have been shown to affect disease manifestation and aggregation of α-synuclein (Appel-Cresswell *et al*, 2013; Hoffman-Zacharska *et al*, 2013; Lesage *et al*, 2013). Moreover, familial PD cases with locus multiplications, such as duplication and triplication of *SNCA*, exhibited severe forms of the disease and early onset (Singleton *et al*, 2003; Chartier-Harlin *et al*, 2004). Duplication or triplication of *SNCA* could produce higher amounts of α-synuclein in neurons, which might account for the aggressive form of the disease observed in those patients (Miller *et al*, 2004; Fuchs *et al*, 2008).

The central role of α-synuclein in the pathogenesis of idiopathic PD was supported by the discovery that aggregates of α-synuclein are major components of Lewy bodies, the proteinaceous intracytoplasmic inclusions in dopaminergic neurons found in post-mortem brains of PD patients (Spillantini *et al*, 1997). In addition, a single-cell study using laser capture microdissection found significantly higher levels of α-synuclein transcripts in PD post-mortem brains (Grundemann *et al*, 2008). Together, these studies indicate regulation of the *SNCA* gene is important and elevated levels of α-synuclein could be a key factor in development of PD and other synucleinopathies.

Numerous studies have investigated the make-up, formation, and propagation of α-synuclein aggregates in PD (Stefanis, 2012; Giraldez-Perez *et al*, 2014); however, little is known about the mechanisms underlying transcriptional or epigenetic deregulation of *SNCA* in PD. Except for two conflicting reports that investigated

---

1 Burnett School of Biomedical Sciences, UCF College of Medicine, University of Central Florida, Orlando, FL, USA
2 Nexmos, Yongin-Si, South Korea
3 Robert Wood Johnson Medical School Institute for Neurological Therapeutics, Rutgers Biomedical and Health Sciences, Piscataway, NJ, USA
  *Corresponding author. Tel: +1 732 235 6499; Fax: +1 732 235 4773; E-mail: yk525@rwjms.rutgers.edu

hypomethylation of the intron 1 CpG island of *SNCA* in PD and control post-mortem brain samples, studies of epigenetic factors regulating the α-synuclein gene in PD have been limited (Guhathakurta *et al*, 2017a; Guhathakurta *et al*, 2017b).

The ENCODE database now allows for in-depth analysis of the epigenetic environment of genes (Consortium EP, 2012). And while the importance of histone post-translational modifications (PTMs) in regulation of gene expression has been established, to date, no studies have comprehensively investigated the potential role of histone PTMs in regulation of *SNCA* in PD (Guhathakurta *et al*, 2017a).

In this study, we investigated the epigenetic environment of *SNCA* with special emphasis on histone PTMs that are potentially enriched in the regulatory region of the gene. Interestingly, we observed a significant increase in one histone PTM, histone H3 lysine 4 trimethylation (H3K4me3) at the *SNCA* promoter in post-mortem PD samples. This transcription-initiating histone mark is one of the major determinants of gene transcription (Barski *et al*, 2007). Reduction of H3K4me3 in neuronal cell lines and PD-derived induced pluripotent stem cell lines (iPSCs) using the dCas9-Suntag system-mediated locus-specific targeting approach decreased levels of α-synuclein. Results of this epigenetic investigation could open new avenues for therapeutic intervention to reduce α-synuclein expression by preventing enrichment of H3K4me3 in the *SNCA* gene.

# Results

### Analysis of histone architecture of human *SNCA* in post-mortem midbrain samples

Human *SNCA* consists of six coding exons and two upstream non-coding exons and spans a 114-kb region in chromosome 4 (Guhathakurta *et al*, 2017a). We first searched *in silico* for any histone marks in the adult human brain substantia nigra (SN) region using the Roadmap Epigenomics Database (http://www.roadmapepigenomics.org/data/tables/adult). In total, the database lists seven histone PTMs in the SN region of the brain from two adult post-mortem samples: H3 lysine 27 trimethylation (H3K27me3), histone H3 lysine 36 trimethylation (H3K36me3), histone H3 lysine 4 monomethylation (H3K4me1), histone H3 lysine 4 trimethylation (H3K4me3), histone H3 lysine 9 trimethylation (H3K9me3), histone H3 lysine 9 acetylation (H3K9ac), and histone H3 lysine 27 acetylation (H3K27ac). However, for *SNCA*, not all seven histone PTMs were enriched in their cohort of tissues analyzed. H3K36me3, the histone mark associated with full-length transcription, was enriched throughout the gene body, ensuring *SNCA* is actively transcribed in the SN region. Most interestingly, we observed that H3K4me3, H3K27ac, and H3K27me3 were the three histone PTMs preferentially enriched in the primary regulatory regions of *SNCA*—the area ranging from approximately −1 kb to +1.5 kb of the transcription start site (TSS). This transcriptionally important region includes the promoter, its upstream regions, and the intron 1 area of the gene (chr4: 90,757,022–90,759,612 bp; GRCh37/hg19). Fig 1A and Appendix Fig S1 shows the distribution pattern of the three histone marks around the promoter region for exon 2, the first coding exon. Both H3K4me3 and H3K27ac are transcription-favoring histone

marks. H3K4me3 is the principal mark associated with transcription initiation, and H3K27ac is an active enhancer-associated mark often enriched at the promoter. By contrast, H3K27me3 is associated with gene repression.

In *SNCA*, we observed both the transcription-promoting marks, H3K4me3 and H3K27ac, had sharp peaks around the TSS and surrounding areas, while H3K27me3 had an overall low-level distribution at the promoter and intron 1 areas. We next analyzed these three histone marks in our cohort of post-mortem midbrain tissue samples of PD and matched controls specifically from the SN region using chromatin immunoprecipitation (ChIP) (Fig 1B and C; Appendix Fig S2). We found that H3K4me3 was significantly enriched at the *SNCA* regulatory region in PD samples compared to controls ($P < 0.0004$; Fig 1B and C). On the other hand, no significant difference was observed for H3K27ac between control and PD subjects (Appendix Fig S2A and B). We also found relatively higher enrichment of H3K27me3 in PD; however, the difference in H3K27me3 between control and PD was less prominent and significant than the difference in H3K4me3 between the two groups (Appendix Fig S2C and D). Enhancer activity of the *SNCA* intron 4 region has been reported (Soldner *et al*, 2016; Guhathakurta *et al*, 2017a). As mentioned above, enhancer regions are often enriched by H3K27ac as shown in the Roadmap epigenetic data for *SNCA* (Soldner *et al*, 2016; Guhathakurta *et al*, 2017a). However, we did not find any significant difference in the enrichment of H3K27ac in the intron 4 enhancer region between control and PD subjects (Appendix Fig S3).

### Correlation between H3K4me3 levels and high levels of α-synuclein in PD patients

We evaluated α-synuclein levels in midbrain SN tissues from control and PD subjects by Western blot analysis. The entire cohort of samples used for analyzing enrichment of H3K4me3 was used for α-synuclein expression. We found that α-synuclein was significantly higher in PD compared to controls ($P < 0.05$; Fig 2A and B). We then compared whether α-synuclein levels could be correlated with corresponding H3K4me3 enrichment. We calculated the median level of α-synuclein among all study subjects (0.43) and found that nine PD subjects and three controls had higher α-synuclein levels than the median. Interestingly, higher levels of α-synuclein in those subjects were significantly correlated with corresponding H3K4me3 enrichment at the gene promoter (Spearman correlation coefficient (R), 0.71, $P = 0.01$; Fig 2C). This further established that enrichment of H3K4me3 at the *SNCA* promoter could account for higher expression of α-synuclein.

### Relative enrichment of H3K4me3 in the neuronal population of post-mortem PD brain

Cellular heterogeneity in the brain is a major obstacle when studying neuronal epigenomic architecture. We further investigated whether the observed difference in H3K4me3 between control and PD is maintained in neuronal populations in the SN region. We isolated 20,000 neuronal nuclei from the SN region of a subset of brain samples (6 control and 7 PD) using NeuN-based fluorescence-activated nuclei sorting (FANS). NeuN antibody is a widely used neuron-specific antibody that preferentially binds to Fox3

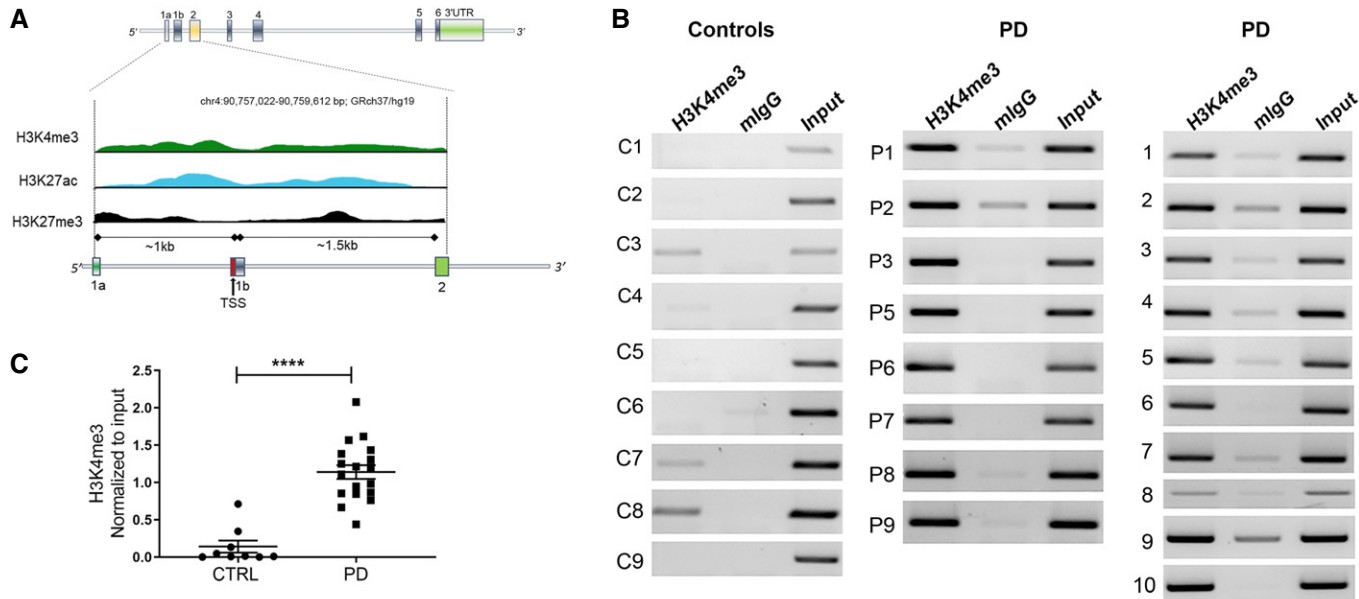

**Figure 1.   Parkinson's disease patients harbor high levels of H3K4me3 at the α-synuclein promoter.**

A   Human *SNCA* gene contains six coding exons and two 5′ non-coding exons. The exons are represented by vertical colored boxes. The region from exon 1a to exon 2 (~2.5 kb) is scaled up to show the distribution of histone PTMs. Distribution of the histone PTMs from the SN region of one donor brain sample is shown. Peaks of three different histone PTMs, H3K4me3 (green), H3K27ac (blue), and H3K27me3 (black), at the regulatory region of *SNCA* were adopted from Roadmap Epigenomics Database. For a detailed view, please see Appendix Fig S1 where screenshot of the original figure is shown. The TSS is indicated by a red vertical bar and distances of exon 1a and 1b from the TSS are indicated.

B   ChIP gel images showing the relative enrichment by H3K4me3 in controls (*n* = 9) and PD patients (*n* = 18). PCR amplified a 188-bp region of *SNCA* from intron 1 where H3K4me3 peak was at its optimum. Mouse IgG (mIgG) was used as control and the bands were normalized by unbiased amplification from respective inputs.

C   Relative intensities calculated from the gel images. Graph shows that H3K4me3 was significantly enriched at the upstream regulatory region of *SNCA* in PD compared to control subjects. ****$P$ < 0.0001. Data were analyzed using non-parametric *t*-test followed by Mann–Whitney *post hoc* corrections. Two-tailed *P*-values were calculated for all.

Data information: Data represent mean ± standard error of the mean.

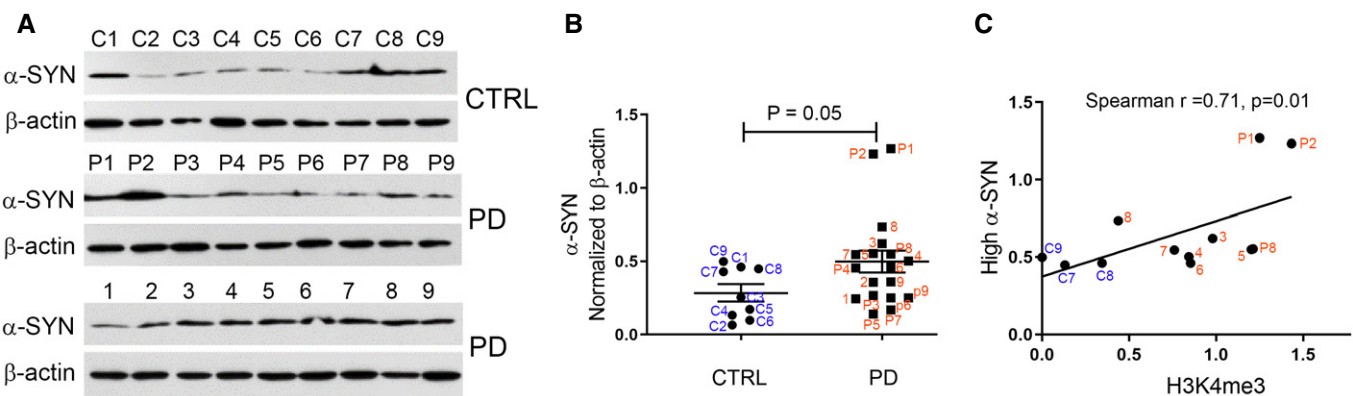

**Figure 2.   PD patients exhibit high expression of α-synuclein.**

A   Western blot gel images showing α-synuclein (α-SYN) levels in SN tissues from control (*n* = 9) and PD subjects (*n* = 18). The numbers on top of each gel panel show the ID of each sample.

B   The relative levels of α-synuclein between the two groups were evaluated statistically. PD subjects had moderately higher levels of α-SYN compared to controls. ID of each dot is shown in the graph.

C   The normalized levels of α-synuclein in the entire cohort were divided into high (*n* = 12) and low levels. The cutoff value for determining the threshold was set at the median from all the subjects. The high levels of α-synuclein were plotted with corresponding H3K4me3 values of those subjects. The corresponding ID of each sample is shown next to each point in the graph. There was a significant correlation between high levels of α-synuclein and H3K4me3.

Data information: Data represent mean ± standard error of the mean. Data in (B) were analyzed using non-parametric *t*-test followed by Mann–Whitney *post hoc* corrections and two-tailed *P*-values were calculated. Spearman rank correlation test was performed in (C). Two-tailed *P*-value was calculated with Fisher's exact *P*.

transcription factor in the neuronal nuclei (Mullen *et al*, 1992; Kim *et al*, 2009). We were able to amplify neuron-specific transcripts (NeuN, synaptophysin, and α-synuclein), but not for astrocytes (GFAP) from the isolated NeuN$^+$ nuclei, confirming the purity of

neuron-specific isolation (Fig 3A and B). As expected, neuronal nuclei isolated from equal amounts of tissue from the SN region yielded much higher numbers of NeuN$^+$ nuclei in controls compared to PD (Fig 3). Using the same primer set used for whole-tissue ChIP

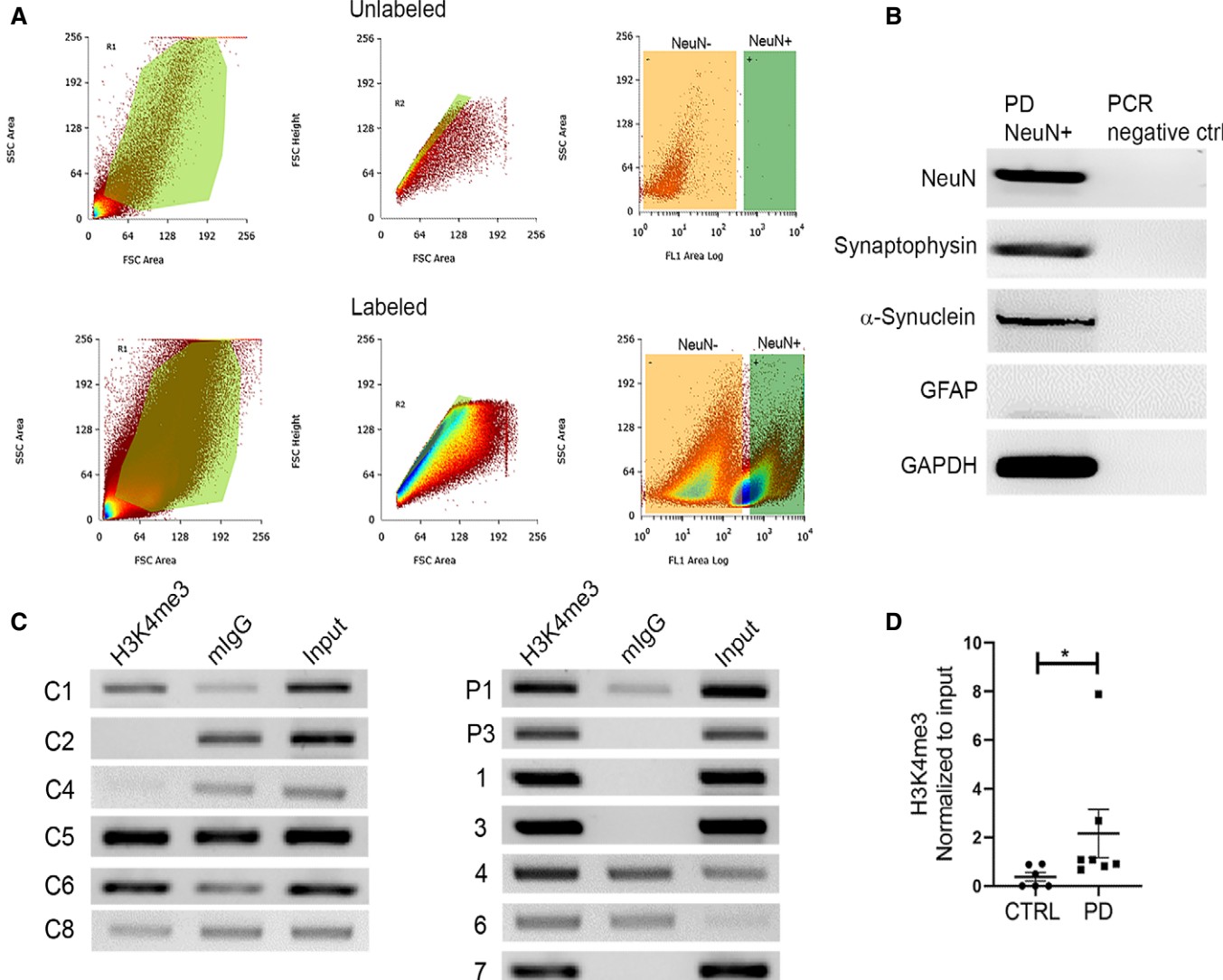

**Figure 3. NeuN-positive neurons from the SN demonstrate high enrichment by H3K4me3.**

A 20,000 NeuN labeled neuronal nuclei were collected by fluorescent activated nuclei sorting (FANS) from control (*n* = 6) and PD (*n* = 7) SN tissues. The NeuN-positive nuclei were labeled by anti-rabbit IgG secondary antibody tagged with Alexa fluor 488. The top and bottom panels show representative sort gating windows from an unlabeled and labeled patient sample, respectively. The left column represents size versus granularity gatings. The samples were then gated for singularity using forward scatter height (*y*-axis) versus forward scatter area (*x*-axis), and lastly, singularly gated nuclei were sorted based on NeuN positivity (*x*-axis; FL1 channel) versus side scatter (*y*-axis). The green rectangular quadrants on the right column of both panels represent the NeuN$^+$ region, as unstained sample did not show any significant representation. Therefore, from each of the stained samples, 20,000 bright NeuN$^+$ nuclei (green rectangle gate) were sorted and collected.

B Gel images from RT–PCR show purity of the sorted nuclei from a representative PD sample. Nuclear RNA was isolated, and cDNA was generated and pre-amplified (see Materials and Methods for details) before target-specific PCR. Neuron-specific genes (NeuN, synaptophysin) and astrocyte-specific gene (GFAP), α-synuclein, and GAPDH were amplified from the isolated nuclei.

C ChIP was performed on the equal number of isolated nuclei against H3K4me3 from all samples. Gel images represent the ChIP-based PCR amplification. The same primer pair was used to amplify the target region on *SNCA* as Fig 1. Mouse IgG was used as a control.

D The graph represents the statistically significant difference of relative H3K4me3 enrichment between PD (*n* = 7) and controls (*n* = 6). Neuronal nuclei from PD brain samples show significantly higher enrichment of H3K4me3 at *SNCA* intron 1 (*P* = 0.01) compared to controls. *$P$ < 0.05. Data were analyzed using non-parametric *t*-test followed by Mann–Whitney *post hoc* corrections. Two-tailed *P*-values were calculated for all.

Data information: Data represent mean ± standard error of the mean.

of H3K4me3 at the *SNCA* promoter/intron 1 region, we found that NeuN[+] nuclei from PD samples demonstrated significantly higher enrichment of H3K4me3 compared to controls ($P = 0.015$; Fig 3C and D).

### Design of CRISPR/dCas9 SunTag-JARID1A system and its recruitment at the *SNCA* promoter

As the histone mark H3K4me3 was significantly enriched at the *SNCA* promoter in PD patients and was correlated with high levels of α-synuclein in all study subjects, we aimed to investigate whether removal of H3K4me3 at the *SNCA* promoter affected α-synuclein levels. To this end, we searched for an epigenetic eraser that could efficiently remove trimethylation at lysine 4 from the histone H3 tail. Histone lysine demethylases (KDM) are specific "erasers" for methylated lysine residues on histones (Hyun *et al*, 2017). KDM5A, also known as JARID1A (Jumonji, AT-rich interactive domain, member 1A), has been shown to have specific demethylating activity at H3K4me3 (Klose *et al*, 2007; Horton *et al*, 2016). JARID1A contains several conserved domains including n- and c-terminal

JmjN and JmjC domains, a DNA binding ARID domain, three PHD domains, a $Zn^{2+}$ binding domain, and a PLU domain (Horton *et al*, 2016). The first 797 amino acids containing JmjN, ARID, PHD1, JmjC, and $Zn^{2+}$ domains were identified as catalytic because overexpression of a recombinant construct containing these domains was sufficient to demethylate H3K4me3 *in vitro* (Horton *et al*, 2016). Based on this information, we selected JARID1A as the H3K4me3 demethylase for our next set of experiments. To recruit a JARID1A catalytic domain to the *SNCA* promoter, we used the CRISPR/dCas9 technology-based SunTag or SuperNova system by replacing the synthetic transcription activator, VP64, with the JARID1A catalytic domain (Fig 4A). Originally, the SunTag system was designed to activate gene expression (up to 300×) by recruiting multiple copies (10 or 22) of VP64 tagged with a single chain variable fragment (scFV) that specifically binds to a GCN4 peptide at the target gene promoter (Tanenbaum *et al*, 2014). Upon co-expression of a dead-Cas9 (dCas9) containing c-terminal repeats of 10 or 22 GCN4 peptides (10× or 22× GCN4) together with genomic locus-specific small guide RNAs (sgRNAs), target gene-specific accumulation of VP64 is achieved, ensuring robust transcriptional activation

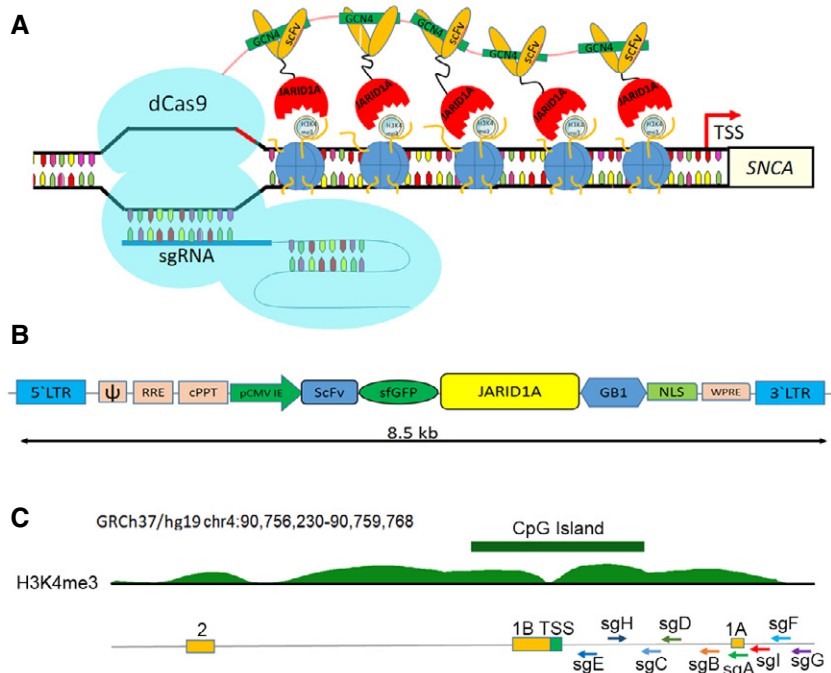

**Figure 4. Design of CRISPR/dCas9 based SunTag-JARID1A system.**

A  Schematic diagram shows how the SunTag-JARID1A system is recruited at the *SNCA* promoter. The dCas9-5xGCN4, scFV-JARID1A, and sgRNA plasmids are co-overexpressed in the cells. The dCas9-5xGCN4 is recruited to the *SNCA* promoter as directed by the specific sgRNA. Five scFV-JARID1A molecules in turn recognize the GCN4 polypeptide sequences of dCas9. Upon recruitment of the entire system, SunTag-JARID1A demethylates H3K4me3 at the target region.

B  Structure of pLvx-scFV-sfGFP-JARID1A. The scFV-sfGFP catalytic domain of JARID1A was sub-cloned into a lentiviral vector. The distance between the 5' and 3' Long Terminal Repeats in the vector is 8.5 kb. The components of the plasmids are as follows: ψ, packaging signal; RRE, rev response element; cPPT, central polypurine tract; pCMV IE, immediate early cytomegalovirus promoter; scFV, single chain variable fragment; sfGFP, super folder GFP; JARID1A; catalytic domain of JARID1A; GB1, solubility tag protein; NLS, nuclear localization signal; WPRE, woodchuck hepatitis virus (WHP) post-transcriptional regulatory element.

C  Relative positions of the nine sgRNAs. A ray diagram exhibits the locations of the nine sgRNAs in relation to the upstream regulatory regions of *SNCA* and the H3K4me3 and H3K27me3 peaks. The locations of the sgRNAs are as follows with respect to the TSS: sgA, 1,117 bp; sgB, 836 bp; sgC, 747 bp; sgD, 700 bp; sgE, 153 bp; sgF, 1,454 bp; sgG, 1,537 bp; sgH, 132 bp; sgI, 1,320 bp. The two non-coding exons (1A and 1B) along with the first coding exon (exon 2) are also shown. GRCh37/hg19 contig was used to describe the location of the gene and associated distribution of histone PTMs. The histone peaks shown were from midbrain regions of two adult post-mortem human samples.

(Tanenbaum *et al*, 2014). We sub-cloned the scFV-sfGFP-JARID1A catalytic domain into a lentiviral system where the transgene is expressed under the CMV immediate enhancer element (Fig 4B). The entire sequence of this novel vector system is provided in Appendix Fig S15. Morita *et al* (2016) previously showed that to avoid steric hindrance between adjacent effector molecules and for proper recruitment of larger effectors such as TET1 catalytic domain, a minimum 22 amino acid spacer is necessary between each GCN4 unit. Therefore, we also used this dCas9-5xGCN4 system, which could recruit five JARID1A molecules at the *SNCA* promoter site when co-overexpressed with scFV-sfGFP-JARID1A and sgRNA (Fig 4A). We designed nine guide RNAs upstream of the TSS of *SNCA* where the peaks of H3K4me3 were relatively higher (Fig 4C). Guide RNA design was carried out using Broad Institute's genomic perturbation platform to ensure selection of only those sequences that had 0 to 1 unintended potential off-targets elsewhere in the genome (see Materials and Methods for details). The list of sgRNAs with sequences and their relative locations are provided in Appendix Table S3.

To determine whether dCas9-5xGCN4 and scFV-sfGFP-JARID1A could form a complex when expressed together, we transiently co-overexpressed these constructs in HEK293 cells and immunoprecipitated using anti-Cas9 antibody and further blotted against anti-GFP antibody (Appendix Fig S4). We observed that cells expressing both dCas9-5xGCN4 and scFV-sfGFP-JARID1A were successfully immunoblotted against anti-GFP antibody followed by Cas9-mediated precipitation, demonstrating that scFV-sfGFP-JARID1A forms a stable complex with dCas9-5xGCN4 when co-expressed (Appendix Fig S4A and B). By contrast, the cells expressing only dCas9-5xGCN4 could not pull down anything else, confirming specificity.

### Significant reduction in α-synuclein expression by removal of H3K4me3 at the *SNCA* promoter

SH-SY5Y cells, a neuronal cell line, exhibit a relatively high level of α-synuclein expression compared to other neuronal cell lines (Appendix Fig S8). Therefore, we chose this cell line to see whether recruitment of JARID1A at the promoter region of *SNCA* could reduce its expression. First, we created stable SH-SY5Y cell lines expressing dCas9-5xGCN4, which was confirmed by PCR using primers spanning the GCN4 regions and by immunofluorescence and immunoblotting against Cas9 protein (Appendix Figs S5 and S7). Using these cells, we further developed cell lines that stably express each sgRNA to recruit the SunTag system precisely at the targeted genomic locations. All nine sgRNA cell lines were able to recruit dCas9-5xGCN4 precisely at distinct locations of the *SNCA* promoter (Appendix Fig S6). Finally, we stably transfected scFV-sfGFP-JARID1A in these cell lines which were already expressing dCas9-5xGCN4 and individual sgRNAs (Appendix Fig S10) (see Materials and Methods for details). Out of nine different sgRNA cell lines, only two sgRNAs, sgA and sgD, successfully reduced H3K4me3 from the *SNCA* promoter (Appendix Fig S9). Between sgA and sgD, sgA was selected for the rest of the study, as it completed removed H3K4me3 from the *SNCA* promoter of SH-SY5Y.

The sgA is located on the non-coding exon 1A, which is approximately 1 kb upstream of the TSS (Fig 4C). SH-SY5Y cells expressing dCas9-5xGCN4/sgA/scFV-sfGFP-JARID1A (dCas9-sgA-JA) exhibited

significantly reduced H3K4me3 (almost to null) at the promoter of the gene ($P < 0.05$) compared to control cells expressing dCas9-5xGCN4/sgA (dCas9-sgA; Fig 5A and Appendix Fig S12A). The significant reduction in H3K4me3 resulted in a notable decrease in expression of α-synuclein (Fig 5B and C and Appendix Fig S12B). To confirm that reduction in α-synuclein was not caused by any of the individual components of the tripartite SunTag system, we separately overexpressed the three components in SH-SY5Y cells and did not observe any significant changes in α-synuclein levels (Appendix Fig S11A and B). Next, we investigated global H3K4me3 levels in cell lines expressing dCas9-sgA-JARID1A compared to wild-type SH-SY5Y cells. We did not observe any significant difference in global H3K4me3 levels between these cells (Appendix Fig S11C).

### Efficient reduction in α-synuclein by CRISPR/dCas9 SunTag-JARID1A in PD patient-derived iPSCs

We previously reported that dopaminergic neurons differentiated from sporadic PD patient-derived iPSC lines (sPD) expressed significantly higher levels of α-synuclein than did neurons derived from control iPSCs (Je *et al*, 2018). In addition, the level of tyrosine hydroxylase (TH) expression did not differ between PD and control iPSC lines when differentiated completely (Je *et al*, 2018). The pluripotency of these lines was validated by immunostaining using four different markers (Appendix Fig S13). Next, we investigated whether JARID1A could reduce α-synuclein levels in these sPD-iPSC lines. As expected, differentiated sPD-iPSC cells exhibited high enrichment of H3K4me3 at the *SNCA* promoter as compared to the control iPSC lines (Fig 6A and C), while no difference in H3K27me3 was observed (Fig 6D). To determine whether locus-specific recruitment of JARID1A could reduce H3K4me3 at the *SNCA* promoter in PD-iPSCs, we selected a sPD1-1 line and transiently transfected sgA with scFV-sfGFP-JARID1A and dCas9-5xGCN4 on day 25 of differentiation. We confirmed that sPD-iPSC lines were adequately differentiated by day 25–30; they exhibited sufficient number of dopaminergic neurons expressing TH (33.96%; Appendix Fig S14). Successful decrease in H3K4me3 at the *SNCA* promoter was achieved after transfection with dCas9-5xGCN4/sgA/scFV-sfGFP-JARID1A compared to control (dCas9-5xGCN4/sgA/scFV-empty) (Fig 6E and Appendix Fig S12C). As expected, we observed significant reduction (average decrease of 56–66%; $P < 0.0001$) in α-synuclein protein levels in differentiated sPD-iPSC lines administered with the CRISPR/dCas9 SunTag-JARID1A system (Fig 6F and Appendix Fig S12D). These results indicate that locus-specific modulation of H3K4me3 at the *SNCA* promoter effectively ameliorated α-synuclein in PD patient-derived iPSC lines.

## Discussion

Properly orchestrated regulation by epigenetic factors ensures successful gene expression; any disparities in epigenetic regulation can lead to faulty transcription. *SNCA*, the gene encoding α-synuclein, has a complex epigenetic environment, and it is reasonable to hypothesize that deregulation could lead to altered gene expression (Guhathakurta *et al*, 2017a). In this study, we performed an in-depth analysis of the epigenetic regulation of *SNCA* in PD. We

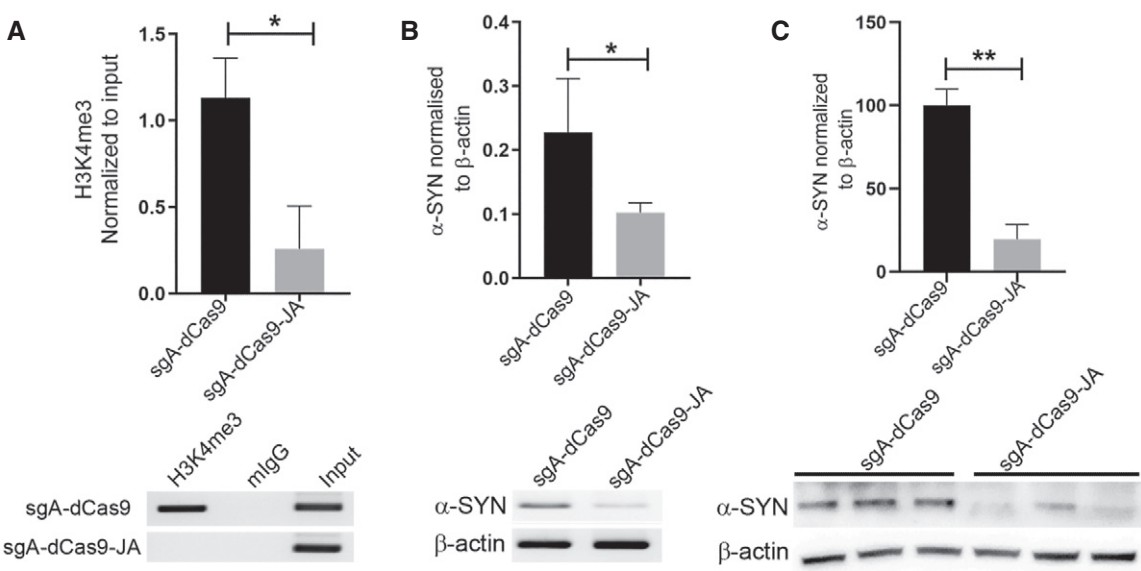

**Figure 5. CRISPR/dCas9 SunTag-JARID1A decreases α-synuclein expression by reducing H3K4me3 at the *SNCA* promoter.**

The relative enrichment of H3K4me3 (A) and corresponding α-synuclein levels (B and C) were evaluated in SH-SY5Y cells expressing dCas9-5xGCN4 with (sgA-dCas9) or without scFV-sfGFP-JARID1A (sgA-dCas9-JA)

A ChIP data demonstrated a significant decrease of H3K4me3 at the *SNCA* promoter. The relative enrichment by H3K4me3 was normalized to the respective inputs. Four independent repeats were performed. *$P < 0.05$. Data were analyzed using non-parametric *t*-test followed by Mann–Whitney *post hoc* corrections. Two-tailed *P*-values were calculated. Only representative gel is shown here, the gels for other replicates are presented in Appendix Fig S12A.

B The levels of α-synuclein (α-SYN) in SH-SY5Y cells were evaluated using RT–PCR. The level of α-SYN was normalized to β-actin expression. A significant reduction of α-SYN in cells expressing scFV-sfGFP-JARID1A was observed. Four independent repeats were performed. *$P < 0.05$. Data were analyzed using non-parametric *t*-test followed by Mann–Whitney *post hoc* corrections. Two-tailed *P*-values were calculated. Only representative gel is shown here, the gels for other replicates are presented in Appendix Fig S12B.

C α-SYN levels in SH-SY5Y cells were also evaluated using Western blot. The levels of α-SYN were normalized to β-actin. Significant reduction of α-SYN was observed in cells expressing JARID1A. Three independent repeats were performed. Percentage of reduction is shown in the graph. **$P < 0.01$. Data were analyzed using non-parametric *t*-test followed by Mann–Whitney *post hoc* corrections. Two-tailed *P*-values were calculated.

Data information: All data are presented as mean ± SEM.
Source data are available online for this figure.

demonstrated that H3K4me3, a histone mark associated with transcription initiation, could serve as a pivotal epigenetic marker for the deregulated expression of *SNCA* in PD. Our data using postmortem brain samples from PD and matched controls clearly indicate that H3K4me3 is significantly enriched at the *SNCA* promoter/regulatory region in PD patients. Successful locus-specific reduction of this histone mark from the *SNCA* promoter reduced levels of α-synuclein significantly in human neuroblastoma, SH-SY5Y, and PD patient-derived iPSC cell lines. This finding could open new research avenues for development of therapeutic strategies for PD and other neurodegenerative diseases by targeting the epigenetic environment of implicated genes.

The promoter and associated regulatory region of human *SNCA* contain several histone PTMs that can regulate α-synuclein's transcriptional state. Three histone PTMs that were enriched at the *SNCA* promoter were H3K4me3, H3K27ac, and H3K27me3. While both H3K27ac and H3K4me3 were present at the *SNCA* promoter, only H3K4me3 was found to be enriched significantly in the patient group compared to controls. We did observe H3K27me3 was significantly enriched in PD group; however, this enrichment was not as significant as H3K4me3. This selective and most significant enrichment of H3K4me3 at the *SNCA* promoter might account

for higher expression of α-synuclein in the SN of PD patients. As mentioned in the methods, we precisely used equal amount of tissues from the SNpc of PD and controls to compare all the histone marks at this locus. Strikingly, this observation of enhanced enrichment of H3K4me3 at this locus indicates that significantly higher proportion of cells had this histone PTM at the *SNCA* promoter in PD compared to the controls, leading to higher expression of α-synuclein in PD.

We subsequently explored another additional histone mark in the same region of the gene, H3K27ac, which favors transcription. Between H3K27ac and H3K27me3 marks, H3K27ac show higher levels of occupancy at the locus in general. However, it did not show any bias toward PD patients. On the other hand, as mentioned previously, H3K27me3 showed a preference toward PD. These relatively higher levels of repressive H3K27me3 and low levels of transcription-favoring H3K27ac at the *SNCA* promoter might counterbalance the effect of H3K4me3 in PD patients. It is obvious that H3K27ac and H3K27me3 compete for the same 27[th] lysine residue on an H3 peptide tail. Therefore, coexistence of these two marks in a single nucleosome is unlikely. Single-cell analysis might be required to precisely determine whether proportions of nucleosomal combination of H3K4me3/H3K27ac are higher

over H3K4me3/H3K27me3 in PD. Also, specific combination of epigenetic architecture could be a cell type-specific event. Previously, we and other groups independently showed that *SNCA* has the intragenic enhancer region represented by the enrichment of H3K27ac in the intron 4 which may contribute to transcriptional activity of this gene (Soldner *et al*, 2016; Guhathakurta *et al*, 2017a). However, we did not observe significant difference of this enhancer marker between the groups. Interestingly, the dinucleotide repeat polymorphism at the NACP-Rep1 locus, which is located around ~10 kb upstream of the TSS, also has been shown to regulate expression activity of *SNCA* (Chiba-Falek & Nussbaum, 2001). The higher repeat alleles were associated with higher activity of the gene and also found to be associated with PD in some populations (Farrer *et al*, 2001; Maraganore *et al*, 2006). In a future study, it would be interesting to see whether the higher repeat alleles of NACP-Rep 1 locus are correlated with enhanced histone PTMs in PD patients.

We also observed α-synuclein protein in PD was higher compared to controls. However, not all the patients exhibited equally high levels of α-synuclein despite having general high H3K4me3 levels at the *SNCA* promoter nor did all the controls exhibit low levels of α-synuclein in spite of having reduced levels of H3K4me3. This also indicates that α-synuclein protein levels are subjected to additional layers of post-transcriptional regulation apart from epigenetic factors. In order to determine whether high levels of

α-synuclein were correlated with H3K4me3 enrichment, we segregated the entire cohort into high and low α-synuclein expressing groups. Overall, high levels of α-synuclein were significantly correlated with higher enrichment of H3K4me3 at the *SNCA* promoter. However, this correlation did not follow complete linearity, suggesting other secondary epigenetic factors could be involved in this complex gene regulation.

Brain tissue has complex architecture comprising different cell types specific to regions of the brain (Guintivano *et al*, 2013). A significant difference in enrichment of H3K4me3 at the *SNCA* promoter/intron 1 region was initially observed using entire tissue sections from the SN. Because the presence of heterogeneous cellular populations in the region can confound results, epigenetic environment can vary by cell type (Guintivano *et al*, 2013), and interpretation of data from the ensemble average of SN tissue for a disease characterized by selective dopaminergic neuronal loss is potentially risky, we further investigated whether the epigenetic changes observed in the entire tissue composition were also present in pure neuronal populations from the SN region. We examined H3K4me3 at the *SNCA* promoter/intron 1 from NeuN[+] neurons isolated from SN tissue of a subset of study subjects. NeuN-FANS yields a homogenous population of nuclei from neurons. Equal number of neuronal nuclei isolated from both groups also showed a significant H3K4me3 enrichment at the *SNCA* promoter in the patient samples, suggesting that SN neurons in PD patients suffer

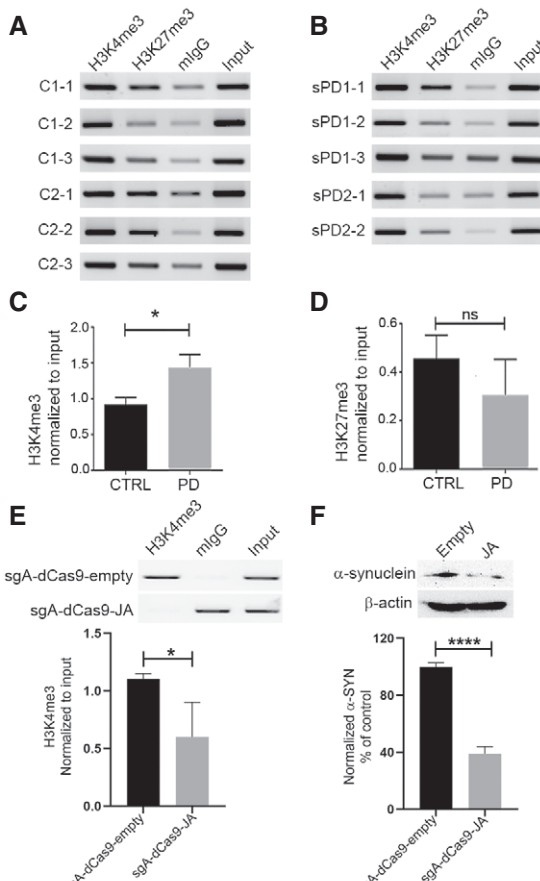

**Figure 6. CRISPR/dCas9 SunTag-JARID1A ameliorates elevated α-synuclein levels in differentiated PD-iPSCs by reducing H3K4me3 from the gene promoter.**

A, B  Relative enrichment of H3K4me3 and H3K27me3 at the *SNCA* promoter/intron 1 region was evaluated from six iPSC lines of two control subjects and five iPSC lines derived from two sporadic PD (sPD) cases by ChIP. The same genomic region as shown in Fig 1 and Appendix Fig S2C was evaluated for enrichment of H3K4me3 and H3K27me3. Mouse IgG (mIgG) was used as control for the target antibody. The relative intensity of the target band was normalized by respective input.

C  ChIP on H3K4me3 at the SNCA promoter between control and sPD1-1 lines revealed a marginally higher enrichment in sPD lines as compared to the controls.

D  Relative comparison of H3K27me3 enrichment between control and sPD lines revealed no significant difference between the two groups.

E  ChIP on H3K4me3 at the SNCA promoter between sPD1-1 lines after locus-specific epigenomic modulation. Differentiated cells were transiently transfected with either sgA-dCas9 5xGCN4-scFV-JARID1A or sg A-dCas9 5xGCN4-scFV-empty backbone vectors. A significant reduction in H3K4me3 at the *SNCA* promoter was observed in cells transfected with JARID1A. A representative ChIP image is shown, and the gels for other replicates are presented in Appendix Fig S12C.

F  Western blot analysis of α-synuclein (α-SYN) levels in the cells under the same conditions as analyzed in (E). A significant decrease (56–66%) in α-SYN levels was observed in cells transfected with JARID1A. The normalized and relative expression of α-SYN in JARID1A transfected cells are shown as a percentage of control (transfected by empty backbone vector). A representative Western blot image is shown at the top, the gels for other replicates are presented in Appendix Fig S12D.

Data information: Three independent repeats were performed for both ChIP and Western blot experiments. *$P < 0.05$; ****$P < 0.0001$. Data were analyzed using non-parametric *t*-test followed by Mann–Whitney *post hoc* corrections. One-tailed *P*-values were calculated for ChIP analysis, while two-tailed *P*-value was calculated for Western blot experiments. All data are presented as mean ± SEM.

Source data are available online for this figure.

from a significant imbalance in epigenetic regulation of α-synuclein. A technical limitation, however, made it difficult to investigate epigenetic changes selectively in dopaminergic neurons. Interestingly, the obvious significant difference in enrichment of H3K4me3 between the patients and controls that was observed from the whole SNpc tissue was slightly reduced when we did the same experiment from the enriched neuronal population, albeit remain significant. This indicates that non-neuronal cells also contributed to this epigenetic deregulation of *SNCA*. It is also possible that slightly high levels of H3K4me3 at the *SNCA* promoter in neurons of controls were partially masked by the non-neuronal population in whole-tissue assay, which become apparent upon investigating the neuronal populations by FANS-ChIP. It is known that astrocyte and other glial cells scavenge extracellular α-synuclein released from neurons (Lindstrom *et al*, 2017; Loria *et al*, 2017). It can be hypothesized that these α-synuclein aggregates are taken up by astrocytes and epigenetically upregulate transcription of its endogenous *SNCA* gene in PD, potentially contributing to higher enrichment of H3K4me3 when total SNpc tissue samples in PD were investigated. Therefore, a future study is warranted to investigate the epigenetic regulation of α-synuclein in astrocytes from PD post-mortem brain samples.

Next, we aimed to determine whether locus-specific removal of H3K4me3 could reduce pathological protein levels. To achieve selective reduction in H3K4me3 enrichment from the *SNCA* promoter, we made a genomic locus-specific molecular tool employing a unique H3K4me3 erasing enzyme, JARID1A. Locus-specific epigenomic editing using the dCas9-CRISPR technology has been shown to be efficacious and powerful in changing transcriptional states of target genes. Several different models have been developed, and the majority of them employ general transcriptional activators, such as VP64, p65, or Rta (or a combination of all three), and synergistic activation mediator (SAM; MS2-p65-HSF1)-based systems to activate gene expression by modulating local chromatin structure (CRISPR activator; Gilbert *et al*, 2013; Tanenbaum *et al*, 2014; Chavez *et al*, 2015; Konermann *et al*, 2015; Chavez *et al*, 2016). Interestingly, it has also been shown that fusing a catalytic core domain of histone acetyl transferase (p300) to dCas9 and targeting it to the enhancer of the target genes could activate transcription of multiple genes (Hilton *et al*, 2015). Simultaneously, transcriptional inhibition systems (CRISPR inhibitor) were developed by recruiting various transcriptional inhibitory proteins such as KRAB (Kruppel-associated box domain of Kox1) or four concentrated copies of mSin3 interaction domain (SID4X) within a few hundred bases downstream of the TSS of the target genes (Gilbert *et al*, 2013; Konermann *et al*, 2013). These systematic genetic interrogation systems have proven extremely beneficial in whole genome screening platforms for identifying disease–phenotype-related genes and genes responsible for survival against toxins and chemotherapeutic drugs (Gilbert *et al*, 2014; Shalem *et al*, 2015). However, only a handful of studies have investigated the beneficial effects of altering the epigenetic environment of disease-implicated genes in disease models using epigenetic writer enzymes (Choudhury *et al*, 2016; Liu *et al*, 2018).

To edit the local epigenetic structure of the *SNCA* promoter most efficiently, we used the CRISPR/dCas9 SunTag system where dCas9 tagged with tandem repeats of an epitope allowed us to bring multiple effector molecules to the target site to enhance its effect (Tanenbaum *et al*, 2014). First, we redesigned the original SunTag system that was developed to recruit 10 to 24 copies of VP64 to the target site using a single guide RNA, achieving significant transcriptional activation of the target genes (Tanenbaum *et al*, 2014). JARID1A enzyme was selected to remove H3K4me3. Previously, Hilton *et al* (2015) showed that using the catalytic core of JARID1A as an epigenetic eraser works more efficiently than using the whole enzyme. Since it was previously shown that first 797 amino acids of JARID1A have the most capacity to remove H3K4me3, we replaced VP64 in the original SunTag system with the catalytic domain of JARID1A. Considering the relatively large size of the JARID1A catalytic domain, to avoid steric hindrance and proper recruitment of this bulky enzyme, we added 22 amino acid spacers in between five GCN4 tails at the c-terminus of dCas9 (Morita *et al*, 2016). Our modified system was able to reduce significantly enrichment of H3K4me3 at the *SNCA* promoter when stably overexpressed in SH-SY5Y cells. For the target region on the *SNCA* promoter/intron, we were not restricted to recruit effectors to −200 to −400 bp of the TSS since we were not modulating gene expression directly using transcriptional activators or inhibitors. Instead, we designed guide RNAs based on the distribution of H3K4me3 peaks upstream of the TSS in *SNCA* (from −132 bp to −1,537 bp). We observed that in SH SY5Y cells, targeting the area around 1 kb upstream of the TSS most significantly reduced H3K4me3 from the promoter. As shown in Fig 5B, the reduction of H3K4me3 from those sites in *SNCA* significantly reduced α-synuclein levels, indicating that H3K4me3 is one of the principal epigenetic regulators of the gene. CRISPR/dCas9 SunTag-JARID1A is a tripartite system requiring all three components, sgRNA, dCas9-5xGCN4, and scFV-sfGFP-JARID1A, to ensure removal of H3K4me3 mark from the target site. Overexpression of any of the individual components did not result in any change in α-synuclein levels, demonstrating that the effect on α-synuclein was due to focal recruitment of SunTag-JARID1A at the *SNCA* promoter. Minimal off-target reduction of H3K4me3 was observed; we did not find any changes in global H3K4me3 levels in the genome when we compared relative levels of global H3K4me3 between wild-type cells with sgA-dCas9-JARID1A line. Previously, Morita *et al* (2016) reported that the dCas9-based SunTag system with a TET1 catalytic domain as an effector molecule had exceptionally low off-target effects using whole genome bisulfite sequencing. Moreover, when Kantor *et al* (2018) also targeted dCas9-DNMT3A at the *SNCA* intron 1 to increase methylation, they did not find any significant change in global DNA methylation.

After ensuring that the CRISPR/dCas9 SunTag-JARID1A system could efficiently reduce H3K4me3 from the *SNCA* promoter without perturbing epigenetic architecture elsewhere in the genome, we studied its effect on idiopathic PD-derived iPSC lines. As we reported previously, differentiated dopaminergic neurons from these lines exhibited high levels of α-synuclein compared to control lines, which correlated well with the H3K4me3 levels at the gene promoter. We next investigated whether the locus-specific approach was similarly effective in reducing α-synuclein in PD-iPSC lines after differentiation. Since iPSC lines are known to suppress any transgene promoter activity, it was extremely difficult to make stable cell lines expressing all components of the CRISPR/dCas9 SunTag-JARID1A system (Norrman *et al*, 2010). Since the entire system is bigger than the lentiviral genome, we could not use an "all-in-one" lentiviral system. Instead, we adopted an

efficient transient transfection method using magnetofection. This magnet-based co-transfection ensures high transfection efficiency in primary dopaminergic neurons (Underhill et al, 2014). We also observed a good transfection rate in differentiated dopaminergic neurons derived from the PD-iPSC lines. As expected, CRISPR/ dCas9 SunTag-JARID1A again significantly reduced H3K4me3 at the SNCA promoter with a concomitant decrease in α-synuclein levels of around 55–66%. Previously, in an elegantly designed study, Soldner et al (2016) showed that editing a PD-associated point mutation at the intronic enhancer region of SNCA could affect transcription factor binding and transcription efficiency. However, this mutation is rare in the population and it was shown that it could affect expression of α-synuclein up to 1.18 times in cell culture models. Another recent study by Kantor et al (2018) demonstrated significant reduction of α-synuclein by recruiting dCas9-DNMT3A (a single copy of DNMT3A directly fused at the end of dCas9) to SNCA intron 1 in an SNCA-Tri hiPSC line derived from dopaminergic progenitor cells. Although the results were significant, it should be noted that several groups did not find any difference in DNA methylation between PD patients and matched controls (Guhathakurta et al, 2017b). Therefore, increasing methylation at SNCA intron 1 might not be physiologically relevant. Moreover, as reported previously (Gilbert et al, 2013), recruiting dCas9-DNMT3A downstream to the TSS itself might directly hinder transcription by disrupting progress of polymerase along the gene. In our study, we avoided this issue by targeting the region upstream of the principal TSS of the gene where H3K4me3 was found to be enriched.

In sum, our study describes a novel and efficient method of modulating deregulated expression of α-synuclein in PD. Despite its extreme importance, not many studies have investigated epigenetic control of SNCA in neuronal cells or dopaminergic neurons. With advanced molecular techniques like CRISPR/Cas9 and advanced knowledge of tissue-specific epigenetic environment of all genes, it is now possible to edit the local genetic and epigenetic microenvironment of genes implicated in diseases. Here, we reported significant epigenetic deregulation of the SNCA gene in PD and a novel strategy that efficiently ameliorated expression of α-synuclein by editing pathological epigenetic marks in differentiated iPSCs derived from idiopathic PD patients. It is plausible that H3K4me3 at the TSS is equally important for basal transcriptional activity of the SNCA gene both in controls and PD, but the enhanced enrichment of this histone PTM is disease-specific, which may serve as an epigenetic marker for PD. The study indicates a potential therapeutic application in PD by epigenetic modulation of H3K4me3 at SNCA, which might ameliorate α-synuclein-mediated degenerative changes in the disease. It is already proven by other group that knockdown of α-synuclein in idiopathic PD-iPSC-derived neurons rescued them from lysosomal dysfunctions which is commonly associated with neurodegenerative pathologies (Mazzulli et al, 2016). Additionally, in rat model of PD, it was shown that α-synuclein knockdown prevents neurodegeneration significantly (Zharikov et al, 2015). Therefore, we anticipate that our SunTag-JA-mediated epigenetic modulation could be proven useful in rescuing or preventing neurons from degenerative changes as seen in PD. However, future studies are necessary to confirm whether reduction of α-synuclein by epigenetic editing could rescue degeneration in PD using animal models or iPSC-based models of PD.

## Materials and Methods

### Brain tissues

The study involving human post-mortem brain samples was conformed to the principle set out in the Helsinki Declaration of the World Medical Association and the Department of Health and Human Services Belmont Report. All human post-mortem brain samples were obtained from NIH Neurobiobank as mentioned in our previous publication (Guhathakurta et al, 2017b). In this study, we have increased our patient cohort from eight samples (as previously reported (Guhathakurta et al, 2017b)) to 19 samples. All samples were ethnicity (non-Hispanic or Latino), race (white), age-, and sex-matched. Ages ranged from 70 to 89 years for the PD cohort and 54 to 89 years for controls. Post-mortem interval (PMI) ranged from 10.1 to 35.42 h in PD and 6.62 to 30.25 h in controls. The PMIs and the age ranges did not vary significantly between groups. Details of post-mortem brain samples are provided in Appendix Table S1. The Braak staging (Braak et al, 2003) information for the patients was only available for the cohort from McLean Hospital, Harvard Medical School, which is also included in the table. Samples for any assay were chosen randomly. Chromatin immunoprecipitation for H3K4me3 from whole SNpc tissue and Western blot assay to determine the α-synuclein protein were conducted at the same time involving maximum number of samples available. Based on the results, we calculated the minimum number of samples necessary for the rest of the assays using power analysis calculator GPower3.1, indicating that a minimum of six samples per assay with a statistical power of 80% and significance level of 0.05%. Based on that, we randomly selected samples for each assay.

### Antibodies

The following antibodies were used for Western blot analysis: α-synuclein (BD Transduction Laboratories, Clone 42/α-Synuclein (RUO), 610787; dilution 1:500); β-actin (Sigma, A5316; dilution 1:20,000); Flag-tag (Sigma, F3165; dilution 1:5,000); and Horseradish peroxidase (HRP)-conjugated secondary (goat anti-mouse IgG, Jackson Laboratory, 115-035-146; goat anti-rabbit IgG, Jackson Laboratory, 111-035-144; 1:5,000 dilution). For FACS analysis, anti-NeuN antibody was used (EMD Millipore, ABN78; 1:500). For chromatin immunoprecipitation experiments, the following primary antibodies were used: (Normal mouse IgG, Millipore, 12-371; H3K4me3, Abcam, ab8580; H3K27me3, Active motif, 39155; H3K27ac, Abcam, ab4729). Anti-Cas9 antibody (Takara, 632607; 1:1,000) and anti-GFP antibody (Fisher Scientific, MS1315P0; 1:5,000) were used for immunoprecipitation experiments. For immunocytochemistry, TUJ1 (Neuromics, MO15013; 1:500), TH (Santa Cruz, SC-25269; 1:200) and Cas9 (Takara, 632607; 1:500) antibodies were used.

### Cell culture

SH-SY5Y cells were purchased from ATCC and maintained as per their guidelines. Briefly, cells were grown in DMEM/F12 medium (Thermo Scientific, SH30023FS) containing 10% FBS. The HEK293T cells were cultured in DMEM/high glucose medium (Thermo Scientific, SH30243FS) supplemented with 10% fetal bovine serum (FBS)

(Atlanta Biologicals, S10350H). The cells were maintained in a humidified atmosphere of 5% $CO_2$ at 37°C. Cells were seeded at a density of $1.2 \times 10^6$ cells/10 cm dish, $7.5 \times 10^5$ cells/well of 6-well plates, and $0.5 \times 10^5$ cells/well of 24-well plates.

## iPSC culture and differentiation of PD-iPSC derived dopaminergic neurons

Five iPSC lines derived from two PD patients used in the study were provided by Dr. Hanseok Ko at the Johns Hopkins University School of Medicine. The pluripotency of iPSC cell lines was analyzed by immunostaining for different markers including SOX2, Tra-1-60, SSEA4, and OCT4 (Appendix Fig S13). They were differentiated into dopaminergic neurons following a previously published protocol with some modifications (Kriks *et al*, 2011). PD-iPS cells were expanded on Matrigel in mTeSR1 media (Stemcell Technologies, 85850) with doxycycline (1 μg/ml, Sigma, D9891; Chang *et al*, 2014). Cells were dissociated into a single-cell suspension using accutase and seeded at a density of $1.0 \times 10^6$ cells per well on Matrigel-coated 6-well plates in mTeSR1 supplemented with 10 mM ROCK inhibitor Y-27632 (Tocris, 1254).

For differentiation, cells were exposed to 100 nM LDN193189 (Reprocell, 04-0074) from 0~10 days, 10-μM SB431542 (Reprocell, 04-0010) from days 0–6, 100 ng/ml recombinant human sonic hedgehog (SHH) (R&D systems, 464-SH-025CF), 2 μM purmorphamine (Reprocell, 04-0009), and 100 ng/ml recombinant human/mouse FGF-8b (PeproTech, 100-25) from days 1–6 and 3-μM CHIR99021 (Reprocell, 04-0004) from days 3–12. Cells were grown for 11 days in knockout serum replacement medium (KSR; Gibco, 10828-028) containing advanced DMEM/F12 (Gibco, 12-634-010), 20% KSR, 2 mM L-glutamax (Gibco, 35050-061), and 10 mM β-mercaptoethanol (Gibco, 21985-023). The ratio of KSR medium to N2 medium started at 75:25% and was gradually changed to 25:75% starting from day 5 of differentiation to day 10. On day 11 of differentiation, media was changed to B27 media containing Neurobasal media (Gibco, 21103-049), B27 serum supplement (Gibco, 17504-044), and L-glutamax supplemented with CHIR (3 μM), 20 ng/ml BDNF (PeproTech, 450-02), 20 ng/ml GDNF (PeproTech, 450-10-50 μg), 0.2 mM ascorbic acid (Fisher Scientific, A61-100), 1 ng/ml TGFb3 (R&D Systems, 243B3002/CF), and 0.1 mM dibutyryl cAMP (Sigma, D0627-250MG). From day 13 to 20, the same media composition was used except CHIR was added.

On day 20, cells were dissociated using accutase and replated under high cell density conditions ($1 \times 10^5$–$3 \times 10^5$ cells per cm$^2$) on dishes pre-coated with 15 μg/ml polyornithine, 5 μg/ml laminin, and 2 μg/ml fibronectin in differentiation medium for 10~15 days using day 20 media changed alternating days.

## Plasmid constructs

All template plasmids were sourced from Addgene. Specifically, the pCAG-dCas9-5xPlat2AflD plasmid (Addgene #82560) was a gift from Izuho Hatada (Morita *et al*, 2016). The lentiGuide-puro (Addgene #52963) and pHRdSV40-scFv-GCN4-sfGFP-VP64-GB1-NLS (Addgene #60910) plasmids were gifts from Feng Zhang (Sanjana *et al*, 2014) and Ron Vale (Tanenbaum *et al*, 2014), respectively. The catalytically active domains of JARID1A enzyme (1-797 amino acids) were amplified from pcDNA3/HA-FLAG-RBP2 plasmid (Addgene

#14800), a gift from William Kaelin (Klose *et al*, 2007). The amplified fragment containing linker sequences along with *RsrII* sites at both ends was then sub-cloned into pHRdSV40-scFv-GCN4-sfGFP-VP64-GB1-NLS plasmid after removing the VP64 fragment by restriction digestion using *RsrII*. To generate the empty backbone vector, the pHRdSV40-scFv-GCN4-sfGFP-VP64-GB1-NLS plasmid was digested with *RsrII* enzyme and ligated back to compatible ends of the restriction enzyme. The backbone vector produced this way retained the entire promoter—scFV, sfGFP, NLS, and GB1 components (only VP64 was absent). To generate lentivirus from these constructs, all of them were transferred to pLvx-DsRed vector backbone (Clontech). The entire ScFv-sfGFP-JARID1A-GB1-NLS sequence was sub-cloned into pLvx-DsRed vector using *XmaI/SexAI* restriction sites. Using this strategy, we removed the DsRed, PGK promoter, and puromycin from the pLvx vector and replaced in frame with scFV-JARID1A. Importantly, the original Addgene plasmid (#60910) was also a lentiviral vector, but we sub-cloned the insert into pLvx-DsRed vector because the Addgene vector lacked a WPRE sequence, which resulted in low transduction efficiency. Similarly, dCas9-5X GCN4 was also sub-cloned into pLvx-DsRed vector at *XmaI/NotI* restriction sites.

All guide RNAs targeting the *SNCA* promoter were cloned into LentiGuide-puro vectors following a previously described protocol (Sanjana *et al*, 2014). The guide RNAs were designed using the Broad Institute genetic perturbation platform (https://portals.broad institute.org/gpp/ public/analysis-tools/sgrna-design) and were reverified using the CHOPCHOP program (https://chopchop.cbu. uib.no/). To increase specificity and reduce off-target effects, at least three mismatches were allowed at the 3' end before the PAM sequence.

## Transfection

All transient transfection was carried out using magnetofection protocol by Oz Biosciences using their NeuroMag reagent. Briefly, differentiated iPSC cells were grown in a 12-well plate and the media was changed on the day of transfection with 1 ml of antibiotic-free media. The cells were transfected first on day 25 of differentiation. Around 3 μg of total plasmid DNA was transfected per well following equal molar mass for all three vectors (0.16 pmols each). The DNA was then diluted in 150 μl of Opti-MEM media and spun down. The diluted DNA was mixed gently with 9 μl NeuroMag reagent in a different tube and allowed to incubate for 30 min before adding to the cells. After adding the transfection mix to the cells, the plate was put on a magnetic bar overnight in the 37°C incubator, and the next day, the magnet was removed underneath the plate. This process ensured a very high transfection rate without any visible cell mortality. Transgene expression was visualized by expression of sfGFP associated with the SunTag system. To increase transfection efficiency, cells were transfected again the same way after 2 days of the first transfection and harvested 6 days after first transfection.

## Establishing stable SH-SY5Y cell lines expressing the SunTag system

To obtain a pure enrichment by sgRNA and dCas9, cells were treated serially with lentiviral particles containing dCas9 5xGCN4

and sgRNA. The sgRNA vector was resistant to puromycin and dCas9 was resistant to blasticidin. To generate high titer lentiviral particles from the three components of the SunTag system, the individual transfer plasmids were mixed with lentiviral packaging plasmids, psPAX2 and pMD2.G, at a 1:1:1 molar ratio using X-fect transfection reagent in HEK293T cells. Around 48 h post-transfection, the medium containing lentiviral particles was filtered through 0.45 μM PES filter and mixed with lenti X concentrator at a ratio of 1:4. The solution was then incubated at 4°C for a minimum of 4 h and centrifuged for 45 min at 1,500 $g$ to pellet down the viral particles. The pellet was then resuspended in an appropriate amount of sterile phosphate-buffered saline (PBS) to bring the final concentration to 100×. To generate SH-SY5Y cells containing dCas9-5xGCN4, 10 μl concentrated viral particles mixed with 4 μg/ml polybrene (Sigma) was added to the SH-SY5Y cells grown on a 24-well plate. After 48 h of treatment with lentiviral particles, the cells were positively selected under antibiotic blasticidin (5 μg/ml) (Acros Organics, 227420100) for 4 days. The cells that grew under antibiotic selection were expanded and received a second lentiviral transduction for sgRNA. Cells were transduced with 10 μl sgRNA containing lentiviral particles and were allowed to grow for another 2 days before selection with puromycin (2 μg/ml) for 48 h. The puromycin-resistant cells were expanded for 1 week. Cells were then treated with JARID1A containing lentiviral particles for 72 h. Finally, to obtain a pure population of cells containing all three transgene expressions, cells were FAC sorted for GFP (conferred by JARID1A/backbone plasmid). Sorted cells were then grown for another 2 days before harvesting for biochemical analysis.

## Fluorescence-activated nuclei sorting (FANS)

Fluorescence-activated nuclei sorting from post-mortem brain samples were conducted following published protocols with some modifications (Jiang et al, 2008). Briefly, the frozen post-mortem brain tissue (100 mg) around the SN region was dissected on a cryotome. The tissue was then transferred to the glass container of Dounce homogenizer, which already contained 5 ml Nuclear Extraction Buffer (NEB; 0.32 M Sucrose, 5 mM $CaCl_2$, 3 mM $Mg(Ac)_2$, 0.1 mM EDTA, 10 mM Tris–HCl (pH8), 1× Protease inhibitor cocktail, 0.1% Triton X-100) with 1% formaldehyde. Once the tissue was thawed in NEB, it was homogenized 21 times with loose pestle and nine times with tight pestle for about 1 min (Hu et al, 2017). After homogenization, 5 ml of sample solution was transferred to a 15-ml tube at RT for 10 min for optimum crosslinking. Excess formaldehyde was quenched by adding 500 μl 125 mM glycine for 5 min. The homogenized sample was then pelleted by centrifuging it at 2,000 $g$ for 5 min. The supernatant was discarded, and the pellet was resuspended in 5 ml fresh NEB supplemented with protease inhibitor (PI). The resuspended nuclei were then transferred to 38.5-ml ultracentrifuge tube on ice (Beckman Coulter, 344058). To the nuclei suspension, 25 ml of sucrose cushion buffer (1.8 M Sucrose, 3 mM $Mg(Ac)_2$, 10 mM Tris–HCl; pH 8.0) was added to the bottom of the tube. Any volume make-up to the top of the tube was done using NEB. The samples were then centrifuged at 107,163.6 $g$ for 2.5 h at 4°C in an ultracentrifuge (XPN 100, Beckman Coulter). After centrifugation, the supernatant was removed from the tubes on ice and 1 ml pre-chilled PBS was added to the

nuclei pellet. The tubes were incubated for 20 min on ice without any disturbance, and the pellet was resuspended optimally. The sample was then transferred to a tube and mixed with nuclei resuspension buffer (NRB: 250 mM sucrose, 25 mM KCl, 5 mM $MgCl_2$, 20 mM Tri-Cl, 1.0%. BSA, pH 7.4; Supplemented with PI) at 1:2 ratio. The nuclei were then recovered after 5 min centrifuge at 2,000 $g$ and again resuspended in 1 ml NRB on ice. The integrity of the nuclei and number was checked under the microscope from a small aliquot of the resuspended nuclei. From that 1 ml of nuclei suspension, 100 μl suspension was aliquoted in a different tube to make appropriate negative controls such as unlabeled or unstained controls. The volumes were brought to 1 ml with NRB. To the remaining 900 μl nuclei, 1 μl rabbit anti-NeuN antibody was added and the volume was brought to 1 ml with NRB. All the tubes were then incubated overnight at 4°C on a rotating platform. The next day, nuclei were recovered by spinning down at 2,000 $g$ for 5 min and washed twice for 10 min with NRB before incubating with secondary antibody. To the washed sample, 1 μl goat anti-Rb IgG Alexa Fluor-488 secondary antibody was added and incubated for 1 h on a rotating platform at RT in the dark. Next, samples were again washed twice with NRB for 10 min each before FAC sorting. Finally, the samples for FANS were resuspended in PBS + PI and strained through 40-μM nylon filter right before sorting. FANS gatings were made based on a previously published protocol (Jiang et al, 2008). Samples prepared for chromatin immunoprecipitation (ChIP) reactions were directly collected in 350 μl SDS lysis buffer (composition is mentioned under ChIP protocol). A Biorad 2 laser S3e sorter was used to sort the samples.

## Chromatin Immunoprecipitation (ChIP)

Chromatin Immunoprecipitation was performed following the protocol for EZ ChIP™ Chromatin Immunoprecipitation kit (Millipore, 17-371) with the following modifications. The treated or untreated adherent cells (SH-SY5Y and differentiated iPSCs) were fixed by adding required volume of 37% formaldehyde (final concentration 1%) to the medium and incubated for 5 min at RT. The excess formaldehyde was quenched using 125 mM glycine for 10 min. Cells were collected in PBS (pH 7.4) containing PI cocktails following two washes in PBS. For human post-mortem SN tissues, 22 mg of freshly frozen tissue from each sample was isolated by punch biopsy on dry ice and homogenized by Teflon-coated homogenizer pre-cooled in liquid nitrogen. The sample was then cross-linked in 1 ml 1% formaldehyde for 20 min at RT. The supernatant was discarded following a brief spin and 125 mM glycine was added for 5 min to each sample. The samples were then washed twice in 1 ml PBS supplemented with PI. After this step, the tissues were processed as described below. Collected cells/tissues were centrifuged at 400 $g$ for 3 min and resuspended in 350 μl lysis buffer (1% SDS; 10 mM EDTA, pH 8.0; 50 mM Tris-Cl, pH 8.0) for 10 min on ice. Cell suspension was then sonicated in a sonicator (Fisher Scientific, Model No. FB 50) equipped with a probe for microcentrifuge tubes (Model No. CL-18, Fisher Scientific) for five pulses at 20 Hz for 20 s with 30-s intervals in between each pulse. The sonicated cells were centrifuged at 10,000 $g$ for 10 min at 4°C. Remaining supernatant was collected and diluted 10× in ChIP dilution buffer (0.01% SDS; 1.1% Triton X-100; 1.2 mM EDTA, pH 8.0; 16.7 mM Tris-Cl, pH 8.1; 167 mM NaCl) containing

PI. Diluted chromatin was pre-cleared with Salmon Sperm DNA/ Protein A agarose slurry (Millipore, 16-157) at 50 μl slurry/ml of diluted chromatin in an end-to-end rotator overnight at 4°C. The next day in the morning, 30 μl pre-cleared chromatin was kept as input and 900 μl chromatin was incubated with each target antibody (1 μg) at 4°C in a rotor. In addition, 20 μl Salmon Sperm DNA/Protein A agarose slurry was added to each tube after 8 h of incubation with the antibody and left for a combined incubation at 4°C overnight. The next day, the immune complex was retrieved by centrifuging at 4,000 *g* for 1 min and sequentially washed one time in low salt wash buffer (0.1% SDS; 1.0% Triton X-100; 2 mM EDTA, pH 8.0; 20 mM Tris-Cl, pH 8.1; 150 mM NaCl), one time in high salt buffer (0.1% SDS; 1.0% Triton X-100; 2 mM EDTA, pH 8.0; 20 mM Tris-Cl, pH 8.0; 500 mM NaCl), one time in Lithium Chloride (LiCl) wash buffer (250 mM LiCl; 1.0% IGEPAL; 1 mM EDTA, pH 8.0; 10 mM Tris-Cl, pH 8.1; 1.0% Deoxycholic acid), and finally washed twice in Tris-EDTA buffer, pH 8.0 (10 mM Tris-Cl; 1 mM EDTA). The antibody conjugated with fragmented DNA was then eluted by incubating it twice in 0.1 mM NaHCO3; 1.0% SDS buffer for 15 min each. Reverse crosslinking was initiated by adding 200 mM NaCl to the buffer and incubated overnight at 65°C. The next day, it was incubated with 10 μg RNase A (Thermo Fisher Scientific) for 30 min at 37°C to remove RNA contamination. Completion of reverse crosslinking was achieved by adding 10 mM EDTA, pH 8.0; 40 mM Tris-Cl, pH 8.0; 50 μg Proteinase K and incubated at 45°C for 2 h. The fragmented DNA was isolated by phenol: Chloroform: Isoamyl alcohol, DNA pellet was visualized by adding 5–10 μg glycogen and finally dissolved in 20 μl of DNase/RNase-free water. ChIP DNA was then amplified for a segment (188 bp) of *SNCA* intron 1 using primer pair SNCA F3/R3. The sequence is presented in Table S2. The intensity of the band-amplified from mouse IgG control was subtracted from each amplicon from respective target of IP'd DNA and normalized by 1% input from each sample. PCR cycles were kept to 35 for these ChIP-PCR studies to show the maximum difference of *SNCA* promoter-specific enrichment by H3K4me3 between control and PD subjects. Control subjects showed some degree of H3K4me3 enrichment when the PCR cycles were increased to 40 cycles where saturation was observed in input.

For ChIP performed from FANS samples, equal number of GFP+ nuclei were collected in 350 μl SDS lysis buffer supplemented with PI. They were immediately sonicated following previously described parameters. Next, the volume was brought to 2 ml with ChIP dilution buffer, 100 μl protein A/G agarose beads, and PI. The remainder of the process was done as described above.

### Immunocytochemistry

To stain the differentiated dopaminergic neurons from PD-derived iPSC lines, a standard immunocytochemical method was followed (Cristovao *et al*, 2012). Briefly, cells were allowed to differentiate on coated coverslips in a 24-well plate. On day 30 of differentiation, cells were fixed and permeabilized with 4% paraformaldehyde in PBS-T (0.01% tween-20) and incubated overnight at 4°C with primary antibodies (TUJ1, TH) at indicated concentrations. The next day, cells were washed with wash buffer (PBS-T) several times and incubated with appropriate fluorophore labeled secondary antibodies (Alexa fluor 547) at indicated concentrations at RT for 1 h in the

dark. Cells were washed to eliminate non-specific binding and incubated with DNA labeling dye Hoechest132 (20 μM) before mounting on glass slides. Slides were visualized on a Nikon Ti eclipse inverted fluorescence microscope. Stable dCas9-sgRNA SH-SY5Y cells were immunostained with Cas9 antibody. The cells were stained as stated above with only exception of using Alexa fluor 488 as secondary antibody. The iPSC cells were also stained for four different pluripotency markers, Sox2, SSEA, Tra-1-60, and Oct4. They were stained following the protocol outlined in the PSC 4-marker Immunocytochemistry kit by Invitrogen (A24881).

### Western blot

Cells grown on 10-cm dish or 6-well plates were harvested in cold PBS and pelleted down by centrifugation at 600 *g* for 5 min at 4°C. Pelleted cells were subjected to lysis in an appropriate volume of Radio Immuno Precipitation Assay buffer (RIPA; 1% NP-40; 0.5% Sodium Deoxycholate; 0.1% SDS) supplemented with 1× protease and phosphatase inhibitor (Thermo Scientific, 1860932) for 15 min on ice, and supernatant containing proteins were collected following centrifugation at 12,000 *g* for 15 min at 4°C. Approximately 40 μg protein sample was prepared in sample buffer (50 mM Tris–HCl, pH 6.8; 2% SDS; 10% glycerol; 1% β-mercaptoethanol; 12.5 mM EDTA; 0.02% Bromophenol blue) and boiled for 5 min followed by snap chilling on ice. The samples were loaded onto 10% SDS gels. For freshly frozen post-mortem brain tissues, around 20 mg tissue was cut from the SN region and thoroughly homogenized in RIPA buffer supplemented with PI. The protein was extracted by incubating the samples in RIPA buffer overnight with constant rotation at 4°C. After gel electrophoresis, proteins from the gel were transferred to PVDF membrane by cold transfer electrophoresis. To detect α-SYN in human samples, only 15 μg protein per sample was used. Following transfer, the membrane was fixed using 4% formaldehyde and 0.01% glutaraldehyde for 30 min. The membranes were blocked by 5% fat-free milk prepared in Tris-buffered saline (TBS)-Tween 20 (0.1%) for 1 h and incubated overnight with primary antibodies prepared in blocking buffer as mentioned in the text. The membrane was washed three times with wash buffer (TBS-T) and incubated with secondary antibodies for 1 h at RT. The protein bands were visualized by Enhanced Chemiluminescent (ECL) detection reagents (Super Signal West Pico Chemiluminescent Substrate, Thermo Scientific, 34077 or ECL Prime Western Blotting Detection Reagent, Amersham, 45-010-090).

### Total RNA isolation and cDNA synthesis

To measure expression from SH-SY5Y cells, total RNA isolation and consecutive cDNA synthesis were performed as described in our previous publication (Basu *et al*, 2017). The relative level of expression of each gene was normalized by β-actin gene expression. To measure expression from FANS-isolated neuronal nuclei, cDNA preamplification was performed prior to PCR amplification. Briefly, the collected nuclei were pelleted down immediately and lysed in 100 μl RA1 buffer from the Nucleo Spin RNA XS Kit (Takara, 740902.50). This kit is optimized for isolating RNA from as little as one cell. The RNA was eluted according to the protocol in 10 μl nuclease-free water, and equal amounts of RNAs from different samples were immediately subjected to amplification-based cDNA

## The paper explained

### Problem

High levels of α-synuclein protein and its aggregation in the substantia nigra pars compacta (SNpc) of midbrain region are considered as the main culprit for degeneration of dopamine producing neurons in Parkinson's disease. The main problem remains the lack of understanding about the molecular mechanisms how this protein gets expressed more in PD patients and how that initiates misfolding which leads to neurodegeneration. Several hypotheses have been put forward to understand α-synuclein's aggregation behavior. However, there remain caveats in our understanding about dysregulated expression of this protein. Epigenetic regulations are one of the key mechanisms which regulate gene expression. However, how epigenetic factors underlying fine-tuning of α-synuclein expression get deregulated in the disease remains to be elucidated.

### Results

We observed the transcription-promoting histone post-translational modification (PTM), H3K4me3, is exclusively enriched at the α-synuclein gene promoter in post-mortem brain samples of PD patients. The high levels of H3K4me3 at the promoter were positively correlated with higher α-synuclein protein levels in the SN of the brain. We have developed a novel CRISPR/dCas9-based SunTag-JARID1A system which effectively reduced H3K4me3 enrichment from the *SNCA* gene promoter and concomitantly decreased the protein levels both in neuronal SHSY5Y cells as well as in dopaminergic neurons derived from PD patients' iPSCs. These results implicate the importance of H3K4me3 in regulation of α-synuclein in PD.

### Impact

Epigenetic factors confer the first line of regulation for gene expression. Histone PTMs are the most important regulators for tissue-specific gene expression. We identified a histone PTM, H3K4m3, gets deregulated in PD which in turn perturbs α-synuclein expression in the patients. Genomic locus-specific editing of this epigenetic mark in cultured human neurons from patients significantly reduced α-synuclein levels. The impact of this novel approach indicates that H3K4me3 may serve as a molecular target to slow down synucleinopathy-mediated dopaminergic neuronal degeneration in PD.

## Data availability

This study includes no data deposited in external repositories.

**Expanded View** for this article is available online.

## Acknowledgements

Authors gratefully acknowledge NIH Neurobiobank for providing all post-mortem brain samples. Authors also thank Prof. Han-Seok Ko, Johns Hopkins University School of Medicine, for kindly providing all PD patient-derived iPSC lines. Authors sincerely thank Prof. Viviane Labrie, Van Andel Research Institute, Prof. Rajiv Ratan, Burke Medical Research Institute and Prof. Flint M. Beal, Weill Cornell Medicine, for their review and critical scientific comments on the article. Financial support from NIH (1R01NS100919 awarded to YSK) is gratefully acknowledged. Authors also thank Mr. Andrew Knott for language editing, Ms. Annlisa Simon for help with lentivirus generation, bacterial stock preparation, and plasmid DNA isolation and Ms. Anishaa Sivakumar for help with preparation of diagrammatic representation of figures.

## Author contributions

SGT and YSK conceptualized the study. SGT principally executed the work and developed all reagents. SGT and YSK wrote the manuscript. SGT, JK, LA, MKS, SB, GJ, and MBF performed the experiments. EA initially helped with cloning of SunTag vectors. YSK supervised all the work. All authors have carefully read the manuscript and provided their input in editing the manuscript.

## Conflict of interest

The authors declare that they have no conflict of interest.

## For more information

NIH NeuroBioBank Website: https://neurobiobank.nih.gov/

OMIM: https://omim.org/entry/163890

Information on Parkinson's Disease: https://www.parkinson.org/understanding-parkinsons/what-is-parkinsons

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
