## [Review Process File · EMBO Molecular Medicine]

Targeted attenuation of elevated histone marks at SNCA alleviates α -synuclein in Parkinson's disease

Subhrangshu Guhathakurta, Jinil Kim, Levi Adams, Sambuddha Basu, Minkyung Song, Evan Adler, Goun Je, Mariana Fiadeiro, and Yoon-Seong Kim

DOI: [10.15252/emmm.202012188](https://doi.org/10.15252/emmm.202012188)

Corresponding author(s): Yoon-Seong Kim (yk525@rwjms.rutgers.edu)

Review Timeline:

Submission Date:	17th Feb 20
Editorial Decision:	24th Mar 20
Revision Received:	16th Sep 20
Editorial Decision:	16th Nov 20
Revision Received:	30th Nov 20
Accepted:	2nd Dec 20

Editor: Jingyi Hou

Transaction Report:

Thank you for the submission of your manuscript to EMBO Molecular Medicine. We have now received feedback from the three referees who agreed to evaluate your manuscript. As you will see from the reports below, Referees #1 and #3 are more supportive than Referee #2, who is more reserved and raises a series of important issues with regards to the disease relevance of the main findings. The main critical point is about the relatively small effect size for H3K4me3 and that the significance in Fig 2D and 3B seems to be driven by outliers. The second critical point is about the phenotypic consequence of SNCA H3K4me3 that is not sufficiently developed (as commented by both Referees #2 and #3). The referees do offer suggestions to improve and strengthen the conclusions and we would like to ask you to address these comments as indicated.

In particular, during our cross-commenting process (in which the referees are given the chance to make additional comments, including on each other's reports), Referee #2 added:

"1) The most important point is the demonstration that SNCA H3K4me3 occupancy not only regulates aSYN expression a bit, but actually promotes the formation of synucleinopathy. Conversely that targeting the demethylase prevents pathological phenotypes in appropriate model systems. While I understand that this will be most laborious in transgenic mice or in vivo seeding models, phenotypic consequences in the iPS neurons have got to be demonstrated (cf reviewer 3 point 5).

2) I also noted the discrepancy of Fig. 2D and Fig. 3C, where the data distribution for H3K4me3 does not match. Maybe I did not understand the plots properly but the authors must address this issue, as elaborated by reviewer 1 point 3 and related comments.

3) The measurements of aSYN levels are variable indeed and arguably scanning Western blots is not precise enough for these purposes. As repeatedly pointed out by the other referees, qRT-PCR is necessary, here to measure SNCA expression at the RNA level, and for protein I additionally suggest ELISA.

4) The CRISPR/dCas9 SunTag-JARID1a system must be systematically assessed in control and PD iPS derived cells in the attempt to rigorously establish that H3K4me3 within the SNCA locus is a pathogenic event and not one global factor contributing to SNCA expression (cf reviewer 1, point 14; reviewer 3, point 8).

5) The problem of non-normal distribution and outliers in Fig. 2D and Fig. 3B has got to be resolved (cf reviewer 3, point 7 and related comments).

6) I agree with reviewer 1 in terms of ordering and arranging the figures properly.

7) Finally, indeed, reviewer 3 points 1-3 should be addressed, at the very least by some plausible discussion."

Transgenic mice or in vivo seeding models are not mandatory, as this would likely not be feasible in a reasonable timeframe. Phenotypic consequences in the iPS neurons need to be demonstrated, as suggested by both referee #2 and #3. The issues regarding the outliers in Fig.2D and Fig.3B needs to be resolved. All other concerns and comments of the referees need to be convincingly addressed to improve the conclusiveness and clarity.

***** Reviewer's comments *****

Referee #1 (Comments on Novelty/Model System for Author):

The concept studied here is very interesting and the story is presented in a compelling way. The technologies described are very good, and it is clear that the authors worked hard on putting together an articulated piece of work. The manuscript could be of high quality in particular if appropriate revisions completed and comments fully addressed.

Referee #1 (Remarks for Author):

Guhathakurta and colleagues report on the identification of novel epigenetic marks of Parkinson's disease (PD). In particular, they identify a significant enrichment in H3K4me₃ at the SNCA promoter of post-mortem brains of PD patients. Because H3K4me acts as a positive regulator of gene expression, α -synuclein levels are increased. In order to counteract the aberrant increase in the expression of this protein, they develop a CRISPR/dCas9-based demethylating system where the catalytic domain of JARID1A is recruited to the SNCA promoter via targeted sgRNAs. The team shows successful demethylation of the SNCA promoter followed by decreased α -synuclein in neuronal cell lines. In conclusion, they suggest this strategy as a potential therapeutic application for PD in the future.

The study is conceptually very interesting and the story is presented in a compelling way. The technologies described are very good, and it is clear that the authors worked hard on putting together an articulated piece of work. However, the manuscript lacks some important data in support of the reported findings. For instance, controls and characterization experiments are often not included. Because of this, the manuscript can benefit from further revisions and clarifications. Please, refer to the points below for the specifics:

- There are some inconsistencies regarding the data on human material. The Materials and Methods section reports on a patient cohort of 18 samples, but EV Table 1 lists a total of 19 PD samples. PMI starts from 6.62 in PD patients, and not from 10.1. The samples are claimed to be ethnicity-matched in the Materials and Methods section, but no ethnicity information is reported in EV Table 1. More importantly, results for some samples are not reported for all experiments. For instance, why is P4 not shown in Figure 1B and EV Figure 2B? Why is patient 7 the only sample from HMS included in EV Figure 2A and EV Figure 3A?
- On a related note, Figure 2 shows results from a number of patients and controls, but not all the cohorts are taken into consideration. On what basis were the remaining samples excluded from the analysis?
- I understand the rationale behind pooling data from patients and healthy controls. However, it would be interesting to include individual sample IDs in graphs, in order to simultaneously follow the phenotype of each subject. For example, Figure 3A lacks any sort of labels. Also, panel B of the same figure shows that two PD patients seem to have particularly high levels of α -synuclein, and Figure 2D shows one patient with very high levels of H3K4me₃. You show correlation data in Figure 3C, but which of the samples are actually being analysed here? Please, add a sample ID to each dot on the graph.
- In the Results, the authors claim that "all the PCRs were performed for 35 cycles to keep the PCR products in unsaturated conditions". This sentence is more suitable for the Methods section. Besides, this claim is questionable, as 35 cycles typically already denote the non-exponential plateau phase of a PCR reaction.
- The increase in α -synuclein protein levels displayed in Figure 3A/B is only minor for most patients. Authors are advised to soften their claims regarding the detection of "high levels of α -synuclein in all study subjects" in the Results section. In addition, authors should substantiate their results with a complementary method if possible, for instance by analysing SNCA mRNA levels by qRT-PCR.

- As a general comment, please revise the order of figures and panels so as to be chronological. For example, Figure 5B is discussed before Figure 5A, which is quite confusing. In the context of Figure 5, there is no need to include all the independent repeats on the same figure, as they unnecessarily crowd it up. These should be moved to the supplementary data, instead.
- EV Figure 4 is poorly organized. In general, figures should be improved in order to be more consistent in terms of sizes and labels.
- The authors claim that SH-SY5Y cells exhibit "a relatively high level of α -synuclein expression". What other cell lines have been analysed by the authors that contributed to this statement? Please, include a figure for this comparison.
- The authors report to have generated a stable SH-SY5Y cell line expressing dCas9-5xGCN4, for which they provide a genotyping PCR. However, this does not represent absolute evidence for the cell line to be composed of a homogeneous population of cells each expressing the transgene, as a bulk culture with only a fraction of edited cells would show the same result. To overcome this issue, authors should include an immunofluorescence staining for Cas9 to prove constitutive expression in all cells. A western blot for Cas9 is also advised.
- On a similar note, how did the authors confirm stable expression of sgRNAs besides antibiotic selection? How did they ensure that no unedited cell survived the selection? Also, could you please report a map/sequence for the used sgRNA constructs?
- Regarding the generation of the stable scFV-GFP-JARID1A cell line, the authors describe FACS sorting GFP-positive cells. However, they do not report any sorting data. As an additional proof, authors should show immunofluorescence images of GFP-positive cells.
- Figure 5B shows a reduction in α -synuclein expression using RT-PCR but not by western blot. EV Figure 8A shows no reduction in α -synuclein expression using western blot but not by RT-PCR. For consistency, the authors should show the data using both methods for the 2 experiments. Also, EV Figure 8A should include a positive control, i.e. SunTag system-treated cells exhibiting reduced α -synuclein levels. Please, provide quantifications.
- When using iPSCs, it is good practice to report proof of cell type characterization before and after differentiation, even though the cell line has been published before. The authors report immunofluorescence stainings for the obtained dopaminergic neurons, but not for the iPSCs. Also, what was the percentage of the cells are dopaminergic neurons?
- SPD-iPSC-derived neurons are reported to show an enrichment in H3K4me3 at the SNCA promoter. However, Figure 6 does not include any results obtained from healthy iPSC-derived neurons. It is not clear how the authors could talk about enrichment considering the lack of controls for comparison.

Minor comments:

- Typo in the Results section: sgA is located on exon 1A according to Figure 4C, and not on exon 1B.
- Where is the error bar for sgA-dCas9-empty in Figure 6C?
- EV Figure 10: please use a lighter colour to highlight the linkers, as their respective sequences are not visible.

- What was the transfection efficiency of differentiating iPSCs? How was successful transfection assessed? These pieces of information should be included.
- Out of curiosity, were H3K27me3 levels reduced in individuals with increased H3K4me3 and α -synuclein?

Referee #2 (Comments on Novelty/Model System for Author):

The effects strengths for H3K4me3 in PD SN neuronal nuclei (Fig. 2D) and aSYN levels (Fig. 3B) are small and dominated by 1-2 extreme values whereas most data points overlap with controls. More importantly, only the expression of aSYN is determined in PD-derived iPSCs (Fig. 6C) without any assessment of synucleinopathy. Thus, the causative implications for PD pathology are not established by the present circumstantial evidence.

Referee #2 (Remarks for Author):

The issue of epigenetic regulation of the disease-causing aSYN is of major importance. However, the information about histone modifications and expression control in the SNCA locus is extremely limited at present. Therefore, the present study is a timely and important contribution. The authors found small increments of H3K4me3 in the SNCA locus weakly correlating with higher levels of aSYN in the SN of PD patients. While these effects appear statistically significant, they are smallish with large overlap to controls. Thus, it is hard to conclude diagnostic or even causative value. Plaudibly, the authors devoted much efforts to target a demethylase to the identified H3K4me3 site in the SNCA promoter. This novel sophisticated system worked, and the authors could reduce the expression of aSYN in PD iPSCs by more than 50%. That is a remarkable starting point, but to establish disease relevance, the effects on aSYN aggregation and pathology must be addressed in appropriate model systems. Also, maybe I missed it, but did the authors show that CRISPR/dCas9 SunTag-JARID1A normalized SNCA expression specifically in PD cells to control levels? In other words, is this specific histone modification strictly pathological, or does it globally promote aSYN expression also in healthy controls?

Referee #3 (Comments on Novelty/Model System for Author):

The novelty is high and the concept behind the paper very good. However, the data are not as strong in places as they should be. My comments are to try to get the data better so as to better support the novel concept.

Referee #3 (Remarks for Author):

This study by Guhathakurta and colleagues is a very elegant piece of work to first define an epigenetic change in Parkinson's disease post-mortem brain tissue and then to develop a vector-based approach to alter that epigenetic change as a potentially therapeutic approach. My comments are based on looking at the data presented which might be enhanced to better support the excellent concept.

1. Post mortem brain tissue comparison of patient v control material is beset by problems in that the vulnerable dopaminergic neurons you wish to study have died in PD. Therefore, it is remarkable that the increases in H3K4me3 Figs 1B and C are so clear. Does that mean the signal is not coming from dopaminergic neurons, as they are mostly dead and gone? This is really important to consider, and I guess was the rationale for the NeuN work which follows.

2. Related to that is the Fig 2D is much less clear (in just the neurons) than 1C (in all cells). Why might this be? Might the signal in 1C be mostly non-neuronal?

2. There is a lot of variation in the strength of the H3K4me3 signal across patients and controls. Is there any correlation in the chromatin marks with genetic polymorphisms at the promoter, such as GWAS-related SNPs or the Rep1 repeat element? I realise that the study is not powered to be a genetic study, but a brief comment would be interesting.

3. The SNCA expression data (Fig 3) are less clear on PD v control. Might that be as neurons are dieing? The level of neuronal loss will depend on Braak staging. Do the authors have a Braak stage for each patient?

4. The neuronal differentiation period chosen (25-30 days) is very short compared to the more commonly-used timepoints of 50-70 days and so their neurons will likely be very immature. How can the authors be confident to have "mature" neurons? Simply being TH-positive and TUJ1-positive in Supp Fig 9 is no indication of maturity. They may be "adequate" but they are not mature.

5. Do these neurons show a phenotype which might be rescued with a reduction in SNCA?

6. The gel images for the replicate experiments commendably shown in Figure 5 seem quite variable, with the changes in H3K4me3 (on the left) not correlating well with SNCA expression (on the right). Can the authors please comment on this?

7. The significances reported in Fig 2D and Fig 3B seem to driven by one or two outliers, whereas as the rest of the samples (PD v control) seem to lie very close to each other (Fig 2D) or have a very high level of overlap (Fig 3B). How much of this signal is driven by the outliers? Again, how much do the two outliers drive the correlation in Fig 3C?

8. It would help to see the H3K4me3 difference between sporadic PD iPSC dopamine neurons and control iPSC dopamine neurons. Essentially, that would be Figure 1C but replicated for PD v control iPSC dopamine neurons. Then the reader can judge if the vector used in Fig 6C has returned patient SNCA levels back to control SNCA levels. The authors can not say "differentiated dopaminergic neurons from these lines exhibited high levels of α -synuclein compared to control lines, which correlated well with the H3K4me3 levels at the gene promoter" if they do not compare the H3K4me3 difference between sporadic PD iPSC dopamine neurons and control iPSC dopamine neurons. To do this lines from three patients will need to be compared to lines from three controls.

9. Are the neurons healthy after the reduction of SNCA expression shown in Fig 6C? It has been reported that strong acute reduction of SNCA, as opposed to the situation in a *Snca*^{-/-} mouse in which gene loss can be compensated for in development, may be detrimental to neurons.
10. The quantitative data from PCR (eg: Fig 1C) have been calculated from band intensity on gels. Surely, Q-PCR would be a much more accurate method?
11. What is the implication if the mouse IgG (mIgG) band is missing from a sample lane of a gel?

Referee #1:

1. There are some inconsistencies regarding the data on human material. The Materials and Methods section reports on a patient cohort of 18 samples, but EV Table 1 lists a total of 19 PD samples. PMI starts from 6.62 in PD patients, and not from 10.1. The samples are claimed to be ethnicity-matched in the Materials and Methods section, but no ethnicity information is reported in EV Table1.

Response:

Thank you for pointing out the inconsistencies. We have rigorously corrected pointed mistakes in the text. We have now included the race/ethnicity information in the table, in the age/sex column in EV Table 1. PMI has been corrected in the text and sample number discrepancies are corrected as well in the text and tables.

2. More importantly, results for some samples are not reported for all experiments. For instance, why is P4 not shown in Figure 1B and EV Figure 2B? Why is patient 7 the only sample from HMS included in EV Figure 2A and EV Figure 3A?

Response:

P4 was not included in these two particular data describing ChIP for H3K4me3 and H3K27me3 (Fig 1B and EV 2C, D) simply due to the limited amount of tissue left. The amounts of tissue provided by Biobank vary widely from sample to sample and there was not enough of this sample for the ChIP. However, we were able to perform some analysis with this for H3K27ac PTM (EV 2A, B).

As we could find no other studies for our target histone marks in midbrain neurons, we ran power analysis using GPower3.1 software with 80 % of power and 0.05 of significance level to identify how many samples we will need for the next sets of experiment. Our analysis suggested 6 samples would be required and for all subsequent experiments. The samples were chosen randomly without bias for inclusion or exclusion aside from having sufficient sample for each histone mark. We have now included statements about basis of sample selection for each assay using postmortem samples in the "Brain tissues" section under Methods.

3. On a related note, Figure 2 shows results from a number of patients and controls, but not all the cohorts are taken into consideration. On what basis were the remaining samples excluded from the analysis?

Response:

As mentioned earlier, we were limited by practical considerations relating to the amount of sample we were provided. This effect was exacerbated by the fact that in PD, SNpc DA neurons degenerate and more tissue was required to obtain an equal number of neurons for analysis. Based on our previous power analysis, we limited our cohorts to 6 or 7 samples that were chosen randomly with no bias in their selection. For Fig 2D (Fig 3D in the revised manuscript), we included the remaining samples we had by that time after finishing other experiments.

4. I understand the rationale behind pooling data from patients and healthy controls. However, it would be interesting to include individual sample IDs in graphs, in order to simultaneously follow the phenotype of each subject. For example, Figure 3A lacks any sort of labels. Also, panel B of the same figure shows that two PD patients seem to have particularly high levels of α -synuclein, and Figure 2D shows one patient with very high levels of H3K4me3. You show correlation data in Figure 3C, but which of the samples are actually being analyzed here? Please, add a sample ID to each dot on the graph.

Response:

We understand reviewer's concern. We have provided the IDs on the top of the western blot gel pictures in previously assigned Fig 3A and also included IDs in the graphs (Fig 3B, C) to make the figure more understandable to the readers.

Just to clarify, the analysis carried out in Fig 1 and Fig 3 were with the whole tissues (mixed cells populations). Whereas only neurons from the SNs were isolated and used in Fig 2 to draw a conclusion on the PD-specific increase in H3K4me3 and whether that is solely neuronal. The correlation between H3K4me3 and α -synuclein that is shown originated from Fig 3A (α -synuclein level in whole SNpc tissues) and 1B (H3K4me3 enrichment from whole SNpc tissues). For a better flow of the storyline, we now have reordered the Fig 3 as Fig 2 and previous Fig 2 as Fig 3.

5. In the Results, the authors claim that "all the PCRs were performed for 35 cycles to keep the PCR products in unsaturated conditions". This sentence is more suitable for the Methods section. Besides, this claim is questionable, as 35 cycles typically already denote the non-exponential plateau phase of a PCR reaction.

Response:

We have rearranged the sentence and put it in the method section. These PCRs were done from H3K4me3 pull down samples where amount of IP'd DNA as PCR template was very low in comparison to any regular PCR. Usually a PCR band would reach a plateau by 35 cycles if the initial template was enough to start with. In our case, it didn't reach plateau at 35 cycles (we found that even at 42 cycles the products were not saturated) and we felt that this is the best balance where input samples from controls were amplified but no sample was reaching plateau.

6. The increase in α -synuclein protein levels displayed in Figure 3A/B is only minor for most patients. Authors are advised to soften their claims regarding the detection of "high levels of α -synuclein in all study subjects" in the Results section. In addition, authors should substantiate their results with a complementary method if possible, for instance by analyzing SNCA mRNA levels by qRT-PCR.

Response:

We have revised the sentence and softened the claims in this regard. The change is highlighted in the discussion. The significance level in Fig 3B (Fig 2B in the revised manuscript) is 0.05, which is now mentioned in the text and figures in the revised manuscript.

Unfortunately, we do not have RNA samples left from these precious postmortem brain samples and therefore used western blot for this assay. The borderline significant difference of α -synuclein levels is shown clearly.

7. As a general comment, please revise the order of figures and panels so as to be chronological. For example, Figure 5B is discussed before Figure 5A, which is quite confusing. In the context of Figure 5, there is no need to include all the independent repeats on the same figure, as they unnecessarily crowd it up. These should be moved to the supplementary data, instead.

Response:

We have made suggested changes. Please see the revised Fig 5. As per reviewer's suggestion, we have used only representative gels to keep the figure neat. And transferred rest of the gels in EV 13.

8. EV Figure 4 is poorly organized. In general, figures should be improved in order to be more consistent in terms of sizes and labels.

Response:

We have improved the quality of all figures in the revised manuscript.

9. The authors claim that SH-SY5Y cells exhibit "a relatively high level of α -synuclein expression". What other cell lines have been analyzed by the authors that contributed to this statement? Please, include a figure for this comparison.

Response:

Now we included a comparison western blot figure as Supplementary figure (EV 8) between ReNcell VM, LUHMES and SH-SY5Y, showing that SH-SY5Y has higher α -synuclein levels. All these cell lines are human neuronal cells lines. We have modified the sentence in the text as well.

10. The authors report to have generated a stable SH-SY5Y cell line expressing dCas9-5xGCN4, for which they provide a genotyping PCR. However, this does not represent absolute evidence for the cell line to be composed of a homogeneous population of cells each expressing the transgene, as a bulk culture with only a fraction of edited cells would show the same result. To overcome this issue, authors should include an immunofluorescence staining for Cas9 to prove constitutive expression in all cells. A western blot for Cas9 is also advised.

Response:

We included an immunofluorescence picture for dCas9 in stable SH-SY5Y cells and a western blot picture in EV 7A (Cas9 immunofluorescence) and B (Cas9 western blot).

11. On a similar note, how did the authors confirm stable expression of sgRNAs besides antibiotic selection? How did they ensure that no unedited cell survived the selection? Also, could you please report a map/sequence for the used sgRNA constructs?

Response:

In figure EV6, we have shown precise dCas9 localization at the SNCA promoter as recruited by sgRNA using ChIP-PCR, which indirectly proves that those cell lines constitutively express sgRNAs at that location and recruit the entire dCas9 assembly.

It is technically challenging to examine if these small guide RNAs are stably expressed. Instead, we conducted ChIP and western blot experiments in triplicates or quadruplicates originated from different passages and different batches of sgRNA cell lines and got similar results. This indicates that sgRNAs were stably expressed in the cell lines over multiple generations.

The sequences for all sgRNAs are included in the EV table 3. The relative binding location of each sgRNA on the SNCA promoter is shown in the Fig 4C.

12. Regarding the generation of the stable scFV-GFP-JARID1A cell line, the authors describe FACS sorting GFP-positive cells. However, they do not report any sorting data. As an additional proof, authors should show immunofluorescence images of GFP-positive cells.

Response:

We have now included the Immunofluorescence picture of the sgA-dCas9-JA cell line in the figure EV 10.

13. Figure 5B shows a reduction in α -synuclein expression using RT-PCR but not by western blot. EV Figure 8A shows no reduction in α -synuclein expression using western blot but not by RT-PCR. For consistency, the authors should show the data using both methods for the 2 experiments. Also, EV Figure 8A should include a positive control, i.e. SunTag system-treated cells exhibiting reduced α -synuclein levels. Please, provide quantifications.

Response:

We inserted the Western Blot figure along with the RT-PCR data. The new figure is now included as Fig 5C. Similarly, we also added RT-PCR result in EV 11B along with western blot data, EV 11A (EV8 in the original manuscript). All data are quantified and bar graphs with statistical analysis are accordingly provided.

14. When using iPSCs, it is good practice to report proof of cell type characterization before and after differentiation, even though the cell line has been published before. The authors report immunofluorescence staining for the obtained dopaminergic neurons, but not for the iPSCs. Also, what was the percentage of the cells are dopaminergic neurons?

Response:

We have now included characterization of iPSC cells in EV 13 and included suggested information. TH+ neurons were around 33.96 % counted from 10 individual fields from three independent differentiation experiments. We have now included this information in the text.

15. sPD-iPSC-derived neurons are reported to show an enrichment in H3K4me3 at the SNCA promoter. However, Figure 6 does not include any results obtained from healthy iPSC-derived neurons. It is not clear how the authors could talk about enrichment considering the lack of controls for comparison.

Response:

As per reviewer's suggestion, we have included H3K4me3 and H3K27me3 of control iPSC lines as well along with both marks from sPD lines in Fig 6. This also shows a higher enrichment of H3K4me3 in sPD lines as compared to controls. However, the focus of this experiment is to show how SunTag-JARID1A system can reduce α -synuclein level by reducing specific H3K4me3 enrichment from the gene promoter. Our intention was not to show the relative comparison of H3K4me3 between control and PD-iPSC.

These patient-derived iPSC lines had significantly higher levels of α -synuclein ((Je et al., 2018), referred in manuscript) which could serve the best subjects to show the effectiveness of our novel system. Therefore, we selected those lines and reduced the α -synuclein levels using SunTag-JARID1A system as compared to the same PD-iPSC lines without JARID1A (disease control).

Minor comments:

1. Typo in the Results section: sgA is located on exon 1A according to Figure 4C, and not on exon 1B.

Response:

We corrected it in the text.

2. Where is the error bar for sgA-dCas9-empty in Figure 6C?

Response:

We have now recalculated the graph and put the error bar as reviewer suggested.

3. EV Figure 10: please use a lighter color to highlight the linkers, as their respective sequences are not visible.

Response:

Color of the linker is now highlighted with a different color (EV 15).

4. What was the transfection efficiency of differentiating iPSCs? How was successful transfection assessed? These pieces of information should be included.

Response:

We did not measure the transfection efficiency by staining. Transfection efficiency was assessed based on the GFP fluorescence tagged with SunTag-JARID1A. We empirically

estimated the transfection efficiency to be 20-30% every time. To increase the transfection efficiency, we always transfected the cells twice with 2 days interval. We have included this in the text.

5. Out of curiosity, were H3K27me3 levels reduced in individuals with increased H3K4me3 and α -synuclein?

Response:

We did not find such correlation. Patients who demonstrated higher H3K4me3 and α -synuclein levels also had slight enrichment by H3K27me3. Almost all patients had similar levels of H3K27me3 irrespective of H3K4me3 or α -synuclein levels.

Referee #2:

Referee #2 (Comments on Novelty/Model System for Author):

Response:

We thank the reviewer for the comments. To simplify our answers, we are dividing reviewer's comments in three parts as follows-

1. The authors found small increments of H3K4me3 in the SNCA locus weakly correlating with higher levels of aSYN in the SN of PD patients. While these effects appear statistically significant, they are smallish with large overlap to controls. Thus, it is hard to conclude diagnostic or even causative value.

Response:

The relative enrichment of H3K4me3 at the SNCA promoter in PD brain is significantly higher as compared to the controls, indicating an association of this epigenetic marker with the disease. We observed a linear correlation between H3K4me3 enrichment and α -synuclein levels between majority of the study subjects, which resulted in strong correlation (Fig 3C). However, a fraction of control population did show high α -synuclein despite having low H3K4me3 enrichment at the SNCA promoter. Conversely, some PD patients show low α -synuclein but high H3K4me3 enrichment, potentially indicating a more complex cell-type specific regulation of the gene expression. As postmortem brain samples consist of a mixture of different cell types including glial cells and neurons it can be difficult to assess which cell types actually contribute to the altered pathology in PD.

It was shown previously that aged controls without any diagnosed neurodegenerative conditions show high levels of α -synuclein (Chu and Kordower, 2007). It indicates that some aged controls may have deregulated α -synuclein expression.

Despite this, the correlation between α -synuclein deregulation and PD is well established and our data suggests that H3K4me3 enrichment at the SNCA promoter contributes to increased α -synuclein levels. Previous works demonstrated lack of DNA methylation in SCNA intron 1 as a causative agent for increased α -synuclein levels in PD with a mere 2% difference (Guhathakurta et al., 2017). In our case, the histone PTM mark correlates well with α -synuclein levels and based on this we consider our results strongly support our hypothesis and make for a strong conclusion.

2. This novel sophisticated system worked, and the authors could reduce the expression of aSYN in PD iPSCs by more than 50%. That is a remarkable starting point, but to establish disease relevance, the effects on aSYN aggregation and pathology must be addressed in appropriate model systems.

Response:

We appreciate that reviewer agrees with the importance of our novel system in managing α -synuclein levels in PD-iPSC. We believe that reducing the level of this protein could potentially help in protecting dopaminergic neurons against synucleinopathy-mediated degeneration. It is important to note that demonstrating a “rescue effect” using this system both *in vivo* or in iPSC platform is extremely challenging in many ways which needs a completely new set of study. Importantly, non-human models including mouse or rats harbor a completely different epigenetic environment around this gene, so we decided using animal models would not help us understand the epigenetics of the human gene. Moreover, no animal sporadic PD models successfully recapitulate synuclein induced neurotoxicity as it is seen in human. Neither humanized mice models are presently available which has human SNCA gene along with its upstream regulatory regions.

Therefore, we chose to use iPSC-based sporadic PD cell model to prove the efficacy of our system. In order to demonstrate whether epigenetic manipulation of SNCA gene in PD would protect the iPSC-derived dopaminergic neurons from α -synuclein aggregation or degeneration, it would need more than 180 to 200 days of culture. As shown by others, 180-200 days old iPSC-derived neurons successfully recapitulate α -synuclein aggregated morphology or any lysosomal or mitochondrial defects (Mazzulli et al., 2016a; Mazzulli et al., 2016b; Burbulla et al., 2017).

Moreover, it is well established and was shown by several groups that knocking down SNCA expression in PD-iPSC derived neurons or even in animal models of PD, would result in neuroprotection from degeneration by recovering them from synucleinopathy mediated injuries (Zharikov et al., 2015; Mazzulli et al., 2016a). Based on these premises, here we showed that our system can reduce pathogenic levels of α -synuclein in sporadic PD patient-derived iPSC dopaminergic neurons, which is one of the most important steps in managing α -synuclein-based PD pathologies and potentially ameliorating α -synuclein aggregation-mediated pathologies in PD.

3. Also, maybe I missed it, but did the authors show that CRISPR/dCas9 SunTag-JARID1A normalized SNCA expression specifically in PD cells to control levels? In other words, is this specific histone modification strictly pathological, or does it globally promote aSYN expression also in healthy controls?

Response:

H3K4me3 is generally associated with any gene transcribing actively and is not, itself, related to any disease condition. However, in this study we found that H3K4me3 is significantly over-enriched at the SNCA promoter in PD patients as compared to the control subjects. The over-enrichment of H3K4me3 at the SNCA promoter in PD might have arisen due to the fact that significantly higher number of cells in PD had this modification in the promoter as compared to the controls when same amount of tissues/cells were analyzed by ChIP. This suggests that

enrichment of this mark at the *SNCA* promoter is contributing to the pathological increase in α -synuclein levels.

In revised Fig 6A we showed that PD-iPSCs had significantly higher level of H3K4me3 as compared to the control-iPSCs. In our previous publication, we also showed that PD-iPSCs express significantly higher α -synuclein compared to the controls. When we treated the PD-iPSC derived DA lines with SunTag-JARID1A, we were successfully able to reduce H3K4me3 significantly which in turn reduced α -synuclein significantly to a level comparable to the controls (Fig 6B-C).

Referee #3:

1. Post mortem brain tissue comparison of patient v control material is beset by problems in that the vulnerable dopaminergic neurons you wish to study have died in PD. Therefore, it is remarkable that the increases in H3K4me3 Figs 1B and C are so clear. Does that mean the signal is not coming from dopaminergic neurons, as they are mostly dead and gone? This is really important to consider, and I guess was the rationale for the NeuN work which follows.

2. Related to that is the Fig 2D is much less clear (in just the neurons) than 1C (in all cells). Why might this be? Might the signal in 1C be mostly non-neuronal?

Response:

We thank the reviewer for the deep insight into the observations in Fig 1 and 2. We agree that in PD patients most of the dopaminergic neurons had died by the time patient's symptoms had appeared. That was our sole purpose to conduct the experiments in Fig 2 to work with limited number of neurons. When we analyzed the NeuN positive cells, we did see a significantly higher H3K4me3 enrichment in neuronal cells of PD patients as compared to the controls, but that level of significance was relatively less than the same when we used entire tissue from the SN in Fig 1. And we definitely agree with the reviewer, that this indicates that not only neurons but also other non-neuronal cells in the SNpc contribute to the upregulation of *SNCA* by epigenetic change. This point is added in the Discussion.

On a different note, as mentioned previously, we have rearranged the figure orders in the revised manuscript. Fig 3 is now labelled as Fig 2 and old Fig 2 has now become Fig 3. We expect this change will help reader to understand that H3K4me3 and high α -synuclein correlation was done both from whole tissue analysis, not from "only neuronal populations".

3. There is a lot of variation in the strength of the H3K4me3 signal across patients and controls. Is there any correlation in the chromatin marks with genetic polymorphisms at the promoter, such as GWAS-related SNPs or the Rep1 repeat element? I realize that the study is not powered to be a genetic study, but a brief comment would be interesting.

Response:

We appreciate reviewer's interesting comment here. Although the study is not a genetic association study, based on the results, we anticipate that the *SNCA* promoter-specific higher enrichment of H3K4me3 in patients is strongly associated with synucleinopathies in PD. It is

important to mention that the dinucleotide repeat polymorphism at NACP-Rep1 locus, which is located around ~10kb upstream of transcription start site, also has been shown to regulate expression activity of SNCA (Chiba-Falek et al., 2001). Higher repeat alleles were associated with higher activity of the gene and also found to be associated with PD in some populations (Farrer et al., 2001; Maraganore et al., 2006; Appel-Cresswell et al., 2013). In a future study, it would be interesting to see whether higher repeat alleles of NACP-Rep 1 locus are correlated with enhanced histone PTMs in PD patients. As suggested, this point is added in discussion. Presently, however, we consider this is out of the scope of the study since we haven't done any polymorphism studies on these samples.

4. The SNCA expression data (Fig 3) are less clear on PD v control. Might that be as neurons are dying? The level of neuronal loss will depend on Braak staging. Do the authors have a Braak stage for each patient?

Response:

We agree with reviewer. We have Braak staging information for only one of the patient cohorts from HMS/NIH neurobiobank (n=10). The detailed information is now included in the EV Table 1. We want to mention here that, we did not find correlation between Braak staging and α -synuclein protein levels for the said cohort.

5. The neuronal differentiation period chosen (25-30 days) is very short compared to the more commonly used timepoints of 50-70 days and so their neurons will likely be very immature. How can the authors be confident to have "mature" neurons? Simply being TH-positive and TUJ1-positive in Supp Fig 9 is no indication of maturity. They may be "adequate" but they are not mature.

Response:

We agree to reviewer's concern. We have changed the sentence to adequately differentiated instead of fully matured neurons. As TH-positive neurons reach to about 34 % with strong α -synuclein expression at the chosen differentiation point, we decided to do our experiment at that time.

6. Do these neurons show a phenotype which might be rescued with a reduction in SNCA?

Response:

In order to see α -synuclein-mediated pathological changes in differentiated iPSC, it may take 180-200 days, as shown by others (Mazzulli et al., 2016a; Mazzulli et al., 2016b; Burbulla et al., 2017).

We have differentiated the neurons till 30 days as explained above. We do not think, however, it was not enough time to develop any α -synuclein-related pathological phenotypes. However, we consider that as our novel system was able to reduce the α -synuclein significantly in these neurons, it may protect them from α -synuclein-induced degenerative changes in the long-time culture condition.

7. The gel images for the replicate experiments commendably shown in Figure 5 seem quite variable, with the changes in H3K4me3 (on the left) not correlating well with SNCA expression (on the right). Can the authors please comment on this?

Response:

We have updated the figure with new western blot data (Fig 5C) to make it more coherent between histone data and protein or RNA levels. We agree that although exact amount of decreased H3K4me3 and levels of α -synuclein protein or RNA may be little disproportionate, we can clearly show that decrease in H3K4me3 leads to significant reduction of α -synuclein expression. This might be attributed to that dynamicity of histone regulation and capturing expression level are not completely synchronized in time points.

8. The significances reported in Fig 2D and Fig 3B seem to driven by one or two outliers, whereas as the rest of the samples (PD v control) seem to lie very close to each other (Fig 2D) or have a very high level of overlap (Fig 3B). How much of this signal is driven by the outliers? Again, how much do the two outliers drive the correlation in Fig 3C?

Response:

We understand reviewer's point. We are answering it in three parts-

Figure 2D (Fig 3D in the revised manuscript).

Even if we remove both two ($p=0.03$) or one ($p=0.02$) very high H3K4me3 values, the statistics still remains significant as shown below.

Graph with 2 points removed, ($p=0.03$).

Graph with 1 point removed, ($p=0.02$).

Therefore, we are including all the samples used in the study in an unbiased manner.

Figure 3B (Fig 2B in the revised manuscript)

We agree with the reviewer here. The PD group has borderline higher α -synuclein levels as compared to the controls. Therefore, we have now shown the exact p value on the graph ($p=0.05$) to make it clearer. Also, as suggested by reviewer 1, we have softened our claim

regarding levels of protein difference and mentioned an increasing trend of α -synuclein protein in PD samples. We have also provided the IDs against each point on both the graphs in Fig 3B, C. In Figure 3 we used entire cohort for the determination of the difference in protein levels between control and PD groups and found the significance is marginal, we chose to keep all the samples in that particular study in an unbiased manner.

Concurrently, since we have included entire cohort for the mean comparison of α -synuclein levels, we had to consider all of the samples in the correlation study. Even if we remove the two high α -synuclein bearing subjects from the study, the trend of correlation remains similar. In conclusion, we would like to state that while the protein levels remain marginally higher in PD patients, it is undeniable that the trend of α -synuclein expression in patient group is higher as compared to the controls. Potentially the significance effect is masked due to heterogeneity of cell types in the tissue collected and lack of dopaminergic neurons in PD patients at the time of death compared to the controls. We have mentioned this aspect in the discussion.

9. It would help to see the H3K4me3 difference between sporadic PD iPSC dopamine neurons and control iPSC dopamine neurons. Essentially, that would be Figure 1C but replicated for PD v control iPSC dopamine neurons. Then the reader can judge if the vector used in Fig 6C has returned patient SNCA levels back to control SNCA levels. The authors cannot say "differentiated dopaminergic neurons from these lines exhibited high levels of α -synuclein compared to control lines, which correlated well with the H3K4me3 levels at the gene promoter" if they do not compare the H3K4me3 difference between sporadic PD iPSC dopamine neurons and control iPSC dopamine neurons. To do this lines from three patients will need to be compared to lines from three controls.

Response:

We understand reviewer's concern. We now have included H3K4me3, H3K27me3 enrichments at the *SNCA* promoter from both control and sPD iPSC-derived neurons in Figure 6. The comparison shows a higher enrichment of H3K4me3 in PD-iPSC derived neurons.

We have rewritten the sentences regarding this (highlighted in the discussion).

The purpose of this figure and study is to show that our SunTag-JARID1A system is capable of reducing the high levels of α -synuclein in pathologic condition such as PD. Previously we demonstrated an increase in α -synuclein in PD-derived iPSC lines as compared to the control iPSC lines. These same PD-iPSC lines were used in this present study to compare the levels of α -synuclein in PD-iPSC lines with and without JARID1A. We consider this approach is better in demonstrating the efficient reduction of α -synuclein levels in patients than using an iPSC line-derived from another individual (control). Our intention was not to demonstrate any difference between control and PD iPSC lines, neither are we claiming that it is a generalized phenomenon. We are only showing that our novel system is capable of reducing α -synuclein in regular neuronal background (Fig 5 with SH-SY5Y) as well as in pathologic cases as demonstrated by iPSC lines, supporting the idea that this novel system could be useful in disease conditions where high levels of α -synuclein might lead to disease.

10. Are the neurons healthy after the reduction of SNCA expression shown in Fig 6C? It has been reported that strong acute reduction of SNCA, as opposed to the situation in a *SncA*^{-/-} mouse in which gene loss can be compensated for in development, may be detrimental to neurons.

Response:

We have used two rounds of transient transfections to introduce SunTag-JARID1A system into the cells 2 days apart and harvested the cells after 5 days of transfection. Therefore, we monitored the cells for around 7 days with SunTag-JARID1A in them. Within the time of investigation, we did not notice any morphological changes in them. Previously another report by also reduced α -synuclein in PD-iPSC derived DA neurons and did not report any morphological changes as well (Kantor et al., 2018).

11. The quantitative data from PCR (eg: Fig 1C) have been calculated from band intensity on gels. Surely, Q-PCR would be a much more accurate method?**Response:**

We agree with the reviewer that q-PCR provides better comparative data between samples with minute difference as it calculates amplification at real time. However, in our experiment, the difference of H3K4me3-mediated enrichment between control and PD subjects is so obvious, we did not feel the necessity of running q-PCR. Additionally, we selected endpoints where the samples were not reaching saturation and repeated our experiments (even PCRs) three times to reach this conclusion. As our data clearly differentiates the enrichment between control and PD we didn't real time PCR to be necessary.

12. What is the implication if the mouse IgG (mIgG) band is missing from a sample lane of a gel?**Response:**

Conventionally mIgG is used as negative control for any ChIP experiments, showing a relative background that a non-specific IgG can pull down from a specific DNA sample. If it does pull down any DNA from the sample, that is usually considered common background amount between all pull down samples. Therefore, we subtract that amount from all enrichments by any specific antibody-mediated pull down from that sample. If there is no band in mIgG in a specific sample, that is usually considered "no background" in that sample.

Reference Cited in Rebuttal

- Appel-Cresswell S, Vilarino-Guell C, Encarnacion M, Sherman H, Yu I, Shah B, Weir D, Thompson C, Szu-Tu C, Trinh J, Aasly JO, Rajput A, Rajput AH, Jon Stoessl A, Farrer MJ (2013) Alpha-synuclein p.H50Q, a novel pathogenic mutation for Parkinson's disease. *Mov Disord* 28:811-813.
- Burbulla LF, Song P, Mazzulli JR, Zampese E, Wong YC, Jeon S, Santos DP, Blanz J, Obermaier CD, Strojny C, Savas JN, Kiskinis E, Zhuang X, Kruger R, Surmeier DJ, Krainc D (2017) Dopamine oxidation mediates mitochondrial and lysosomal dysfunction in Parkinson's disease. *Science* 357:1255-1261.
- Chu Y, Kordower JH (2007) Age-associated increases of alpha-synuclein in monkeys and humans are associated with nigrostriatal dopamine depletion: Is this the target for Parkinson's disease? *Neurobiol Dis* 25:134-149.
- Farrer M, Maraganore DM, Lockhart P, Singleton A, Lesnick TG, de Andrade M, West A, de Silva R, Hardy J, Hernandez D (2001) alpha-Synuclein gene haplotypes are associated with Parkinson's disease. *Hum Mol Genet* 10:1847-1851.
- Guhathakurta S, Evangelista BA, Ghosh S, Basu S, Kim YS (2017) Hypomethylation of intron1 of alpha-synuclein gene does not correlate with Parkinson's disease. *Mol Brain* 10:6.
- Je G, Guhathakurta S, Yun SP, Ko HS, Kim YS (2018) A novel extended form of alpha-synuclein 3'UTR in the human brain. *Mol Brain* 11:29.
- Kantor B, Tagliafierro L, Gu J, Zamora ME, Ilich E, Grenier C, Huang ZY, Murphy S, Chiba-Falek O (2018) Downregulation of SNCA Expression by Targeted Editing of DNA Methylation: A Potential Strategy for Precision Therapy in PD. *Mol Ther* 26:2638-2649.
- Maraganore DM et al. (2006) Collaborative analysis of alpha-synuclein gene promoter variability and Parkinson disease. *JAMA* 296:661-670.
- Mazzulli JR, Zunke F, Isacson O, Studer L, Krainc D (2016a) alpha-Synuclein-induced lysosomal dysfunction occurs through disruptions in protein trafficking in human midbrain synucleinopathy models. *Proc Natl Acad Sci U S A* 113:1931-1936.
- Mazzulli JR, Zunke F, Tsunemi T, Toker NJ, Jeon S, Burbulla LF, Patnaik S, Sidransky E, Maraganore JJ, Sue CM, Krainc D (2016b) Activation of beta-Glucocerebrosidase Reduces Pathological alpha-Synuclein and Restores Lysosomal Function in Parkinson's Patient Midbrain Neurons. *J Neurosci* 36:7693-7706.
- Zharikov AD, Cannon JR, Tapias V, Bai Q, Horowitz MP, Shah V, El Ayadi A, Hastings TG, Greenamyre JT, Burton EA (2015) shRNA targeting alpha-synuclein prevents neurodegeneration in a Parkinson's disease model. *J Clin Invest* 125:2721-2735.

Thank you for the submission of your revised manuscript to EMBO Molecular Medicine. We have now received the enclosed report from one of the three referees who were asked to re-assess it. Unfortunately, we only managed to obtain the report from Reviewer #3. In the interest of time, I prefer to make a decision now rather than further delaying the process. As you will see the referee is now supportive and I am pleased to inform you that we will be able to accept your manuscript pending the following amendments:

***** Reviewer's comments *****

Referee #3 (Remarks for Author):

The authors have sufficiently addressed my comments and this highly innovative paper is now suitable for publication.

2nd Authors' Response to Reviewers**30th Nov 2020**

The authors have made all requested editorial changes.

Accepted**2nd Dec 2020**

We are pleased to inform you that your manuscript is accepted for publication and is now being sent to our publisher to be included in the next available issue of EMBO Molecular Medicine.

YOU MUST COMPLETE ALL CELLS WITH A PINK BACKGROUND ↓
PLEASE NOTE THAT THIS CHECKLIST WILL BE PUBLISHED ALONGSIDE YOUR PAPER

Corresponding Author Name: Yoon-Seong Kim
Journal Submitted to: Embo Molecular Medicine
Manuscript Number: EMM-2020-12188